



# Uncertainty Estimation with Deep Learning for Rainfall–Runoff Modelling

Daniel Klotz[1], Frederik Kratzert[1], Martin Gauch[1], Alden Keefe Sampson[2], Johannes Brandstetter[1], Günter Klambauer[1], Sepp Hochreiter[1], and Grey Nearing[3]

[1]Institute for Machine Learning, Johannes Kepler University Linz, Linz, Austria
[2]Upstream Tech, Natel Energy Inc.; Alameda, CA, USA
[3]Google Research, Mountain View, CA, USA

**Correspondence:** Daniel Klotz (`klotz@ml.jku.at`)

**Abstract.** Deep Learning is becoming an increasingly important way to produce accurate hydrological predictions across a wide range of spatial and temporal scales. Uncertainty estimations are critical for actionable hydrological forecasting, and while standardized community benchmarks are becoming an increasingly important part of hydrological model development and research, similar tools for benchmarking uncertainty estimation are lacking. This contributions demonstrates that accurate
uncertainty predictions can be obtained with Deep Learning. We establish an uncertainty estimation benchmarking procedure and present four Deep Learning baselines. Three baselines are based on Mixture Density Networks and one is based on Monte Carlo dropout. The results indicate that these approaches constitute strong baselines, especially the former ones. Additionally, we provide a post-hoc model analysis to put forward some qualitative understanding of the resulting models. This analysis extends the notion of performance and show that learn nuanced behaviors in different situations.

## 1 Introduction

A growing body of empirical results shows that data-driven models perform well in a variety of environmental modeling tasks (e.g., Hsu et al., 1995; Govindaraju et al., 2000; Abramowitz, 2005; Best et al., 2015; Nearing et al., 2016, 2018). Specifically for rainfall-runoff modeling, approaches based on Long Short-Term Memory networks (LSTM; Hochreiter, 1991; Hochreiter and Schmidhuber, 1997; Gers et al., 1999) have been especially effective (e.g., Kratzert et al., 2019a, b, 2020).

The majority of machine learning (ML) and Deep Learning (DL) rainfall–runoff studies do not provide uncertainty estimates (e.g., Hsu et al., 1995; Kratzert et al., 2019b, 2020; Liu et al., 2020; Feng et al., 2020). However, uncertainty is inherent in all aspects of hydrological modeling and it is generally accepted that our predictions should account for this (Beven, 2016). The hydrological sciences community has put substantial effort into developing methods for providing uncertainty estimations around traditional models, and similar effort is necessary for DL models like LSTMs.

Currently there exists no single, prevailing method for obtaining distributional rainfall–runoff predictions. Many, if not most, methods take a basic approach where a deterministic model is augmented with some uncertainty estimation strategy. This includes, for example, ensemble-based methods, where the idea is to define and sample probability distributions around different model inputs and/or structures (e.g., Li et al., 2017; Demargne et al., 2014; Clark et al., 2016), but also comprises





Bayesian (e.g., Kavetski et al., 2006) or pseudo-Bayesian (e.g., Beven and Binley, 2014) methods, and post-processing methods
(e.g., Shrestha and Solomatine, 2008; Montanari and Koutsoyiannis, 2012), etc. In other words, most classical rainfall–runoff
models do not provide direct estimates of their own predictive uncertainty; instead, such models are used as a part of a larger
framework. There are some exceptions to this, for example methods based on stochastic partial differential equations, which
actually use stochastic models but generally require assigning sampling distributions a priori (e.g., a Wiener process). These are
common, for example, in hydrologic data assimilation (e.g., Reichle et al., 2002). The problem with these types of approaches
is that any distribution that we could possibly assign is necessarily degenerate, resulting in well-known errors and biases in
estimating uncertainty (Beven et al., 2008).

It is possible to fit DL models such that their own representations intrinsically support estimating distributions while account-
ing for strongly nonlinear interactions between model inputs and outputs. In this case, there is no requirement to fall back on
deterministic predictions that would need to be sampled, perturbed, inverted, etc. Several approaches to uncertainty estimation
for DL haven been suggested (e.g., Bishop, 1994; Blundell et al., 2015; Gal and Ghahramani, 2016). Some of them have been
used in the hydrological context. For example, Zhu et al. (2020) tested two strategies for using an LSTM in combination with
Gaussian processes for drought forecasting. In one strategy, the LSTM was used to parameterize a Gaussian process, and in the
second strategy, the LSTM was used as a forecast model with a Gaussian process post-processor. Gal and Ghahramani (2016)
showed that Monte Carlo Dropout (MCD) can be used to intrinsically approximate Gaussian processes with LSTMs, so it is
an open question as to whether explicitly representing the Gaussian process is strictly necessary. Fang et al. (2019) used MCD
for soil moisture modeling and reported a tendency to underestimate uncertainties. They also tested to extend the approach by
accounting for the aleatoric uncertainty by estimating a Gaussian noise term (as proposed by Kendall and Gal, 2017) and found
that the combination was more effective at representing uncertainty.

Our primary goal is to benchmark several methods for uncertainty estimation in rainfall–runoff modeling with DL. We
demonstrate that DL models can produce statistically reliable uncertainty estimates using approaches that are straightforward
to implement. We adapted the LSTM rainfall–runoff models developed by Kratzert et al. (2019b, 2020) with four different
approaches to make distributional predictions. Three of these approaches use Neural Networks to create and mix probability
distributions (Section 2.3.1). The fourth is Monte Carlo Dropout (MCD), which is based on direct sampling from the LSTM
(Section 2.3.2).

The secondary objective of this work is to help advance the state of community model benchmarking infrastructure to
include uncertainty estimation. We struggled with finding suitable benchmarks for the DL uncertainty estimation approaches
explored here. Ad hoc benchmarking and model intercomparison studies are common (e.g., Andréassian et al., 2009; Best
et al., 2015; Kratzert et al., 2019b; Lane et al., 2019; Berthet et al., 2020; Nearing et al., 2018), and while the community has a
(quickly growing) large-sample dataset for benchmarking hydrological *models* (Newman et al., 2017; Kratzert et al., 2019b),
we lack standardized, open procedures for conducting comparative uncertainty estimation studies. Note that from the references
above only Berthet et al. (2020) focused on benchmarking uncertainty estimation strategies, and then only for assessing post-
processing approaches. We previously argued that data-based models provide a meaningful and general benchmark for testing
hypotheses and models (Nearing and Gupta, 2015; Nearing et al., 2020b), and here we develop a set of data-based uncertainty





estimation benchmarks built on a standard, publicly available, large-sample dataset that could be used as a baseline for future

benchmarking studies.

## 2 Data and Methods

As a framework for a benchmarking procedure, we followed the philosophy outlined by Nearing et al. (2018). They recognized three elements of a benchmarking experiment: (i) data, (ii) metrics, and (iii) baselines (criteria). We added a diagnostic part in the form of a post-hoc model examination. The purpose of such an analysis is to (a) examine if the model properties correspond

to our hydrological intuitions, and (b) demonstrate procedures for checking model properties.

Section 2.1 provides an overview of the dataset that we used here. In short, our data came from the Catchment Attributes and MEteorolgoical Large Sample (CAMELS) dataset that is curated by the US National Center for Atmospheric Research. Sect. 2.2 discusses a suite of performance metrics that we used for benchmarking the uncertainty estimation approaches. Sect. 2.3 introduces the different uncertainty estimation approaches that we benchmark and that we propose as data-driven baselines

for future benchmarking studies. We used exclusively data-driven models because they capture the empirically inferrable relationships (e.g., Nearing et al., 2018, 2020b). Lastly, Sect. 2.4 discusses the different experiments of the post-hoc model examination.

### 2.1 Data: The CAMELS Dataset

CAMELS (Newman et al., 2015; Addor et al., 2017) is an openly available dataset that contains basin-averaged daily meteoro-

logical forcings derived from three different gridded data products for 671 basins across the contiguous United States. The 671 CAMELS basins range in size between 4 and 25,000 $\text{km}^2$, and span a range of geological, ecological, and climatic conditions. The original CAMELS dataset includes daily meteorological forcings (precipitation, temperature, short-wave radiation, and humidity) from three different data sources (NLDAS, Maurer, DayMet) for the time period 1980 through 2010, as well as daily streamflow discharge data from the US Geological Survey. CAMELS also includes basin-averaged catchment attributes related

to soil, geology, vegetation, and climate.

We used the same 531 basins from the CAMELS dataset (Figure 1) that were originally chosen for model benchmarking by Newman et al. (2017). This means that all basins from the original 671 with areas greater than $2000\text{km}^2$ or with discrepancies of more than 10% between different methods for calculating basin area were not considered. Since all of the models that we tested here are DL models, we use the terms *training*, *validation*, and *testing* that are standard in the machine learning

community, instead of the terms *calibration* and *validation* that are more common in the hydrology community Klemeš (*sensu* 1986).

### 2.2 Metrics: Benchmarking Evaluation

Benchmarking requires selecting a set of metrics that capture what modellers want to achieve for a particular application. For benchmarking uncertainty estimations, this translates to testing whether the distributional predictions are reliable and have

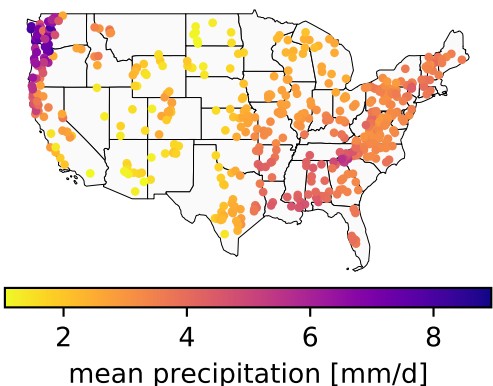

mean precipitation [mm/d]

**Figure 1.** Overview map of the CAMELS basins.

high resolution (terminology adopted from Renard et al., 2010). Reliability measures how consistent the provided uncertainty estimates are with respect to the available observations; and resolution measures the "sharpness" of distributional predictions (i.e., how thin the body of the distribution is). Generally, models with higher resolution are preferable. However, this preference is conditional on the models being reliable. A model should not be overly precise relative to its accuracy (over-confident) or overly disperse relative to its accuracy (under-confident). We note, that the best form metrics for comparing distributional

predictions would be to use proper scoring rules, such as likelihoods (see, e.g., Gneiting and Raftery, 2007). Likelihoods, however do not exist on an absolute scale (it is generally only possible to compare likelihoods between models), which makes these difficult to interpret (although, see: Weijs et al., 2010). Additionally, these can be difficult to compute with certain types of uncertainty estimation approaches, and so are not completely general for future benchmarking studies. We therefore based the assessment of reliability on *probability plots*, and evaluated resolution with a set of *summary statistics*.

**Reliability**. Probability plots (Laio and Tamea, 2007) are based on the following observation: If we insert the observations into the estimated cumulative distribution function, a consistent model will provide a uniform distribution on the interval [0,1]. The probability plot uses this to provide a diagnostic. The theoretical quantiles of this uniform are plotted on the x-axis, and the fraction of observations that fall below the corresponding predictions on the y-axis (Figure 2). Deficiencies appear as deviations from the 1:1 line: a perfect model should capture 10% of the observations below a 10% threshold, 20% under the 20% threshold

and so on. If the relative counts of observations in particular modeled quantiles are higher than the theoretical quantiles this means that a model is under-confident. Similarly, if the relative counts of observations in particular modeled quantiles are lower than the theoretical quantiles then the model is over-confident.

Laio and Tamea (2007) proposed to use the probability plot in a continuous fashion to avoid arbitrary binning. We preferred to use discrete steps in our quantile estimates to avoid falsely reporting overly precise results (e.g., Cole, 2015). As such, we

chose a 10% step size for the binning thresholds in our experiments. We used 10 thresholds in total: one for each of the resulting steps, and the additional 1.0 threshold which used the highest sampled value as an upper bound, so that an intuition regarding



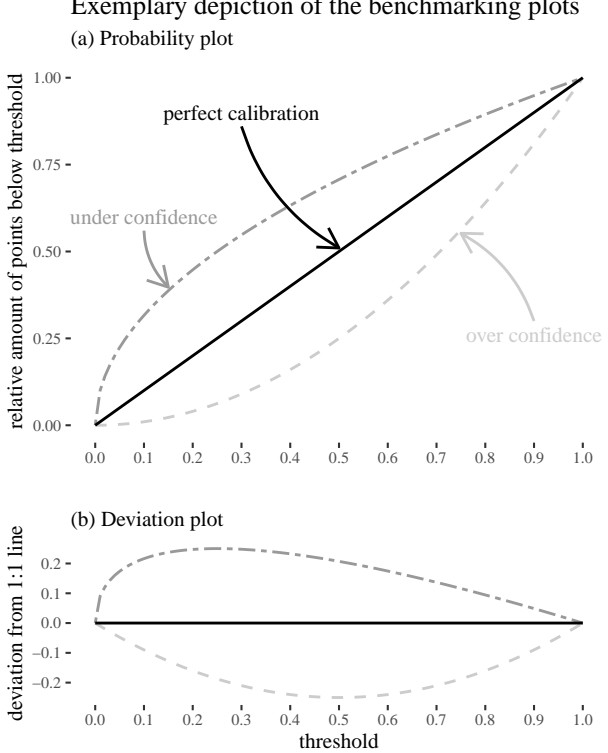

**Figure 2.** Illustration of the probability plot for the evaluation of predictive distributions. The x-axis shows the estimated cumulative distribution over all time steps by a given model, and the y-axis shows the actual observed cumulative probability distribution. A conditional probability distribution was produced by each model for each timestep in each basin. A hypothetically perfect model will have a probability plot that falls on the 1:1 line. We used 10% binning in our actual experiments. The "error plot" (bottom subplot) shows the distances of individual models from the 1:1 line.

the upper limit can be obtained. Subtracting the thresholds from the relative count yields a *deviation from the 1:1 line* (the sum of which is sometimes referred to as *expected calibration error*, see e.g. Naeini et al., 2015). For the evaluation we depicted this counting error alongside the probability plots to provide better readability (see: Figure 2 (b)).

A deficit of the probability plot is its coarseness, since it represents an aggregate over time and basins. As such, it provides a general overview, but necessarily neglects many aspects of hydrological importance. Many expansions of the analytical range are possible. One that suggested itself was to examine the deviations from the 1:1 line for different basins. Therefore, we evaluated the probability plot for each basin specifically, computed the deviations from the 1:1 line and examined their distributions. We did not include the 1.0 threshold for this analysis since it consisted of large spikes only.

**Resolution**. To motivate why further metrics are required on top of the reliability plot it is useful to look at the following observation: There are an infinity of models that produce perfect probability plots. One edge-case example is a model that simply ignores the inputs and produces the unconditional empirical data distribution at every timestep. Another edge-case





**Table 1.** Overview of the benchmarking metrics for assessing model resolution. Each metric is applied to the distributional streamflow predictions at each individual timestep and then aggregated over all time steps and basins. All metrics are defined in the interval $[0, \infty)$ and lower values are preferable (but not unconditional on the reliability).

| Benchmarking metric [a] | Description |
| --- | --- |
| Mean absolute deviation | More robust than standard deviation and variance. |
| Standard deviation | We use Bessel's correction to account for one degree of freedom. |
| Variance | We use Bessel's correction to account for one degree of freedom. |
| Average width of the 0.2 to 0.9 quantiles | We compute the width of each of the inner quantiles and take the mean. |
| Distance between the 0.25 and 0.75 quantiles | Average interquartile range. |
| Distance between 0.1 and 0.9 quantiles | Average interdecile range. |

[a] All metrics are computed for the samples of each timestep and then averaged over time and basins.

example is a hypothetical "perfect" model that produces delta distributions at exactly the observations every time. Both of these models have precision that exactly matches accuracy, and these two models could not be distinguished from each other

using a probability plot. Similarly, a model which is consistently under-confident for low flows can compensate this by being over-confident for higher flows. Thus, to better assess the uncertainty estimations, at least another dimension of the problem has to be checked: the resolution.

To assess the resolution of the provided uncertainty estimates, we used a group of metrics (Table 1). Each metric was computed for all available data points and averaged over all time steps and basins. The results are statistics that characterize

the overall *sharpness* of the provided uncertainty estimates (roughly speaking they give us a notion about how thin the body of the distributional predictions is). Further, to provide an anchor for interpreting the magnitudes of the statistics we also computed them for the observed streamflow values (this yields an unconditional empirical distribution for each basin that can be aggregated). These are not strictly the same, but we believe that they still provide some form of guidance.

## 2.3 Baselines: Uncertainty Estimation with Deep Learning

We tested four strategies for uncertainty estimation with Deep Learning. These strategies fall into two broad categories: Mixture Density Networks (MDN) and Monte Carlo Dropout (MCD). We argue that these approaches represent a useful set of baselines for benchmarking.

### 2.3.1 Mixture Density Networks

The first class of approaches use a Neural Network to mix different probability densities. This class is commonly referred

to as Mixture Density Networks (MDN; Bishop, 1994) and we tested three different forms of MDNs. A *mixture density* is a probability density function created by combining multiple densities, called *components*. An MDN is defined by the parameters of each component and the mixture weights. The mixture components are usually simple distributions like the Gaussians in



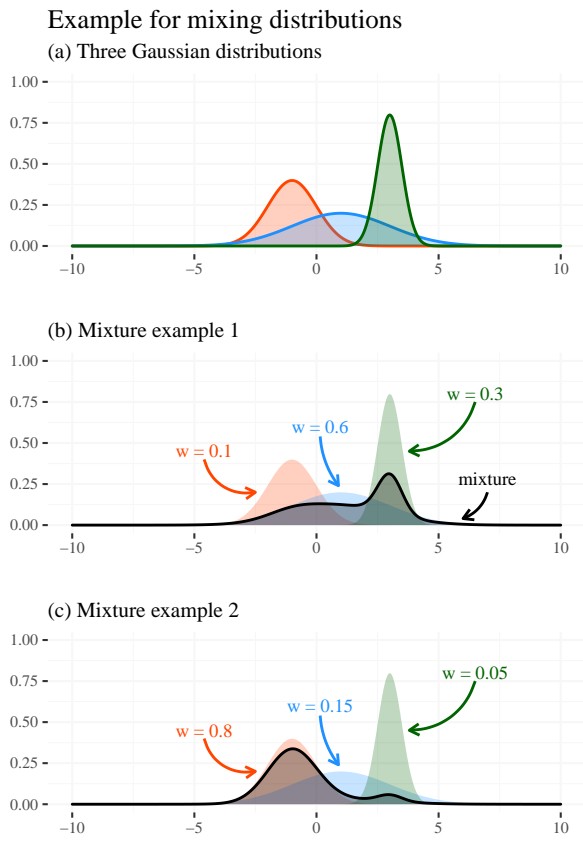

**Figure 3.** Illustration of the concept of a mixture density using Gaussian distributions. Plot (a) shows three Gaussian distributions with different parameters (i.e., different means and standard deviations). Plot (b) shows the same distributions superimposed with the mixture that results from the depicted weighting $w = (0.1, 0.6, 0.3)$. Plot (c) shows the same juxtaposition, but the mixture is derived from a different weighting $w = (0.80, 0.15, 0.05)$. The plots demonstrate that even in the simple example with fixed parameters the skewness and form of the mixed distribution can vary strongly.

Figure 3. Mixing is done using weighted sums. Mixture weights are larger than zero and collectively sum to one to guarantee that the mixture is also a density function. These weights can therefore be seen as the probability of a particular mixture component. Usually, the number of mixture components is discrete, however this is not a strict requirement.

The output of an MDN is an estimation of a conditional density, since the mixture directly depends on a given input (Figure 4). The mixture represents changes every time the network receives new inputs (i.e., in our case for every timestep). We thus obtain time-varying predictive distributions that can approximate a large variety of distributions (they can, for example, account for asymmetric and multimodal properties). The resulting model is trained by maximizing the log-likelihood function of the observations according to the predicted mixture distributions. We view MDNs as *intrinsically distributional*, in the sense that





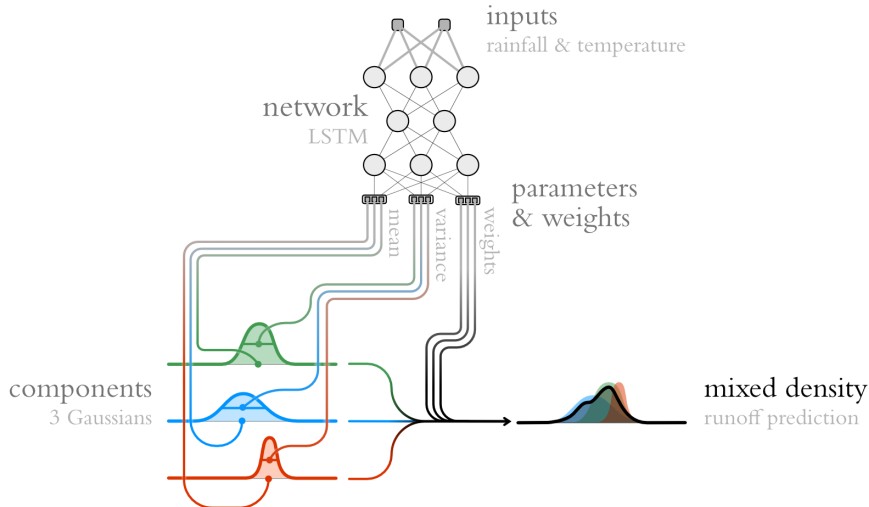

**Figure 4.** Illustration of a Mixture Density Network: The core idea is to use the outputs of a Neural Network to determine the mixture weights and parameters of a mixture of densities (see Figure 3). That is, for a given input, the network determines a conditional density function, which it builds by mixing a set of predefined base-densities (the so-called components).

they provide probability distributions instead of first making deterministic streamflow estimates and then appending a sampling distribution.

In this study, we tested three different MDN approaches:

1. **Gaussian Mixture Models** (GMMs) are MDNs with Gaussian mixture components. Appendix B1 provides a more
formal definition as well as details on the loss/objective function.

2. **Countable Mixtures of Asymmetric Laplacians** (CMAL) are similar to GMMs but instead of Gaussians, the mixture components are asymmetric Laplacian distributions (ALD). This allows for an intrinsic representation of the asymmetric uncertainties that often occur with hydrological variables like streamflow. Appendix B2 provides a more formal description as well as details on the loss/objective function.

3. **Uncountable Mixtures of Asymmetric Laplacians** (UMAL) also use asymmetric Laplacians as mixture components but the mixture is not discretized. Instead, UMAL approximates the conditional density by using Monte Carlo integration over distributions obtained from quantile regression (Brando et al., 2019). Appendix B3 provides a more formal description as well as details on the loss/objective function.

One can read this enumeration as a transition from simple to complex: We start with Gaussian mixture components, then
replace them with ALD mixture components, and lastly transition from a fixed number of mixture components to an implicit approximation. There are two reasons why we argue the more complex MDN methods might be more promising than a simple GMM. First, error distributions in hydrologic simulations often have heavy tails. A Laplace component lends itself towards



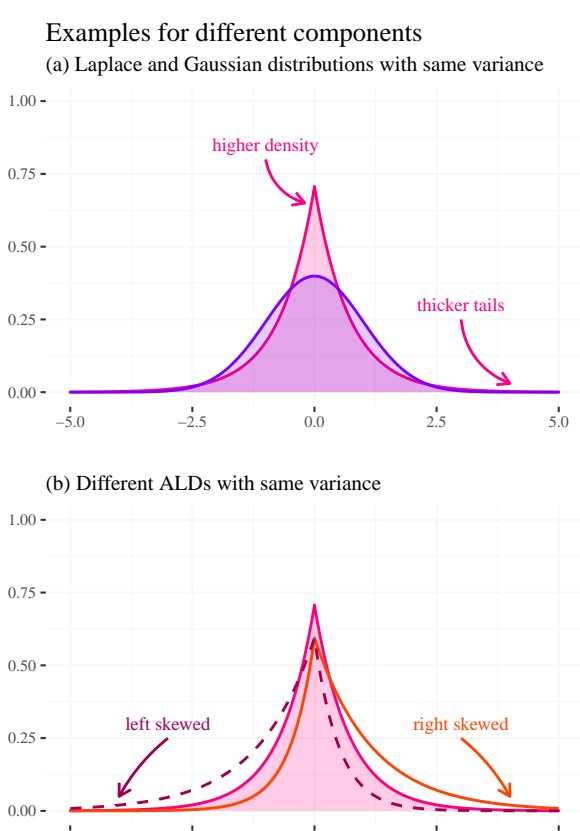

**Figure 5.** Characterization of distributions that are used as mixture components in our networks. Plot (a) superimposes a Gaussian with a Laplace distribution. The latter is sharper around its center, but this sharpness is traded with thicker tails. We can think about it in the following way: the difference in area is moved to the center and the tails of the distributions. Plot (b) illustrates how the asymmetric Laplace distribution (ALD) can accommodate for differences in skewness (via an additional parameter).

thicker-tailed uncertainty (Figure 5). Second, streamflow uncertainty is often asymmetrical, and thus the ALD component could make more sense than a symmetric distribution in this application. For example, even a single ALD component can be used to account for zero flows (compare Figure 5 (b)). UMAL extends this to avoid having to pre-specify the number of mixture components, which removes one of the more subjective degrees of freedom from the model design.

### 2.3.2 Monte Carlo Dropout

MCD provides an approach to estimate a basic form of epistemic uncertainty. In the following we provide the intuition behind its application.





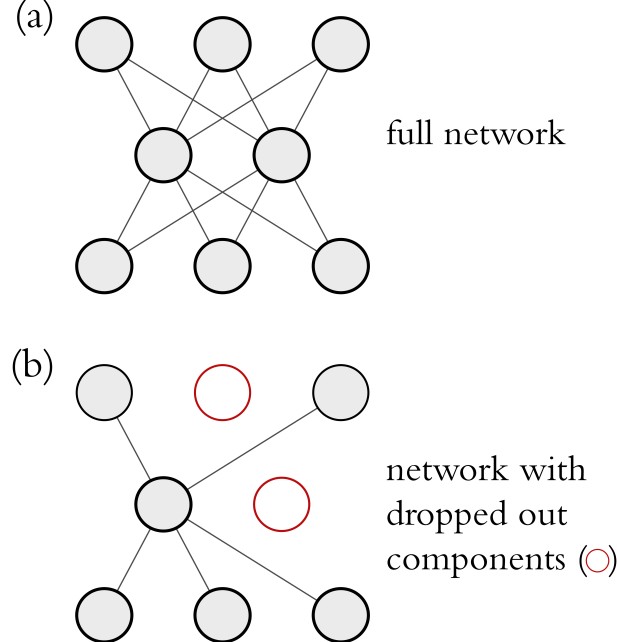

**Figure 6.** Schematic depiction of the dropout concept.

Dropout is a regularization technique for Neural Networks, but can also be used for uncertainty estimation (Gal and Ghahra-mani, 2016). Dropout randomly ignores specific network units (see Figure 6). Hence, each time the model is evaluated during training, the network structure is different. Repeating this procedure many times results in an ensemble of many submodels within the network. Dropout regularization is used during training, while during the model evaluation the whole Neural Network is used. Gal and Ghahramani (2016) showed that dropout can be used as a sampling technique for Bayesian inference –

hence the name *Monte Carlo* dropout.

### 2.3.3   Model Setup

All models are based on the LSTMs from Kratzert et al. (2020) with an additional hidden layer after the LSTM to give the model more flexibility in estimating the mixture weights and components. This means that the presented predictions are from simulation models *sensu* Beven and Young (2013), i.e., no previous discharge observations were used as inputs. The LSTM in

the context of hydrology is inter alia described in (Kratzert et al., 2018), and not repeated here. However, all models can be adapted to a forecasting setting.

In short, our setting was the following: Each model takes a set of meteorological inputs (namely: precipitation, solar radia-tion, min. and max. daily temperature, and vapor pressure) from a set of products (namely: NLDAS, Maurer, and DayMet). As in our previous studies, a set of static attributes is concatenated to the inputs (see: Kratzert et al., 2019b). The training period is

from 01 October 1980 to 30 September 1990. The validation period is from 01 October 1990 to 30 September 1995. Finally,





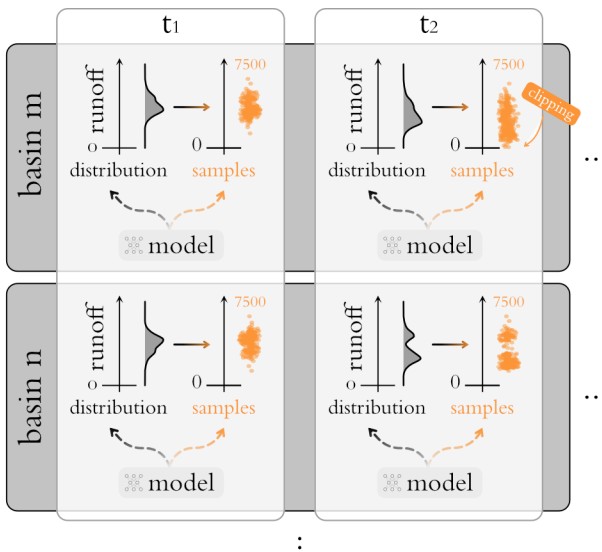

**Figure 7.** Schemata of the general setup. In total we have 531 basin with approximately 3650 data points each. For each time step we compute 7500 samples, by either directly sampling from the model (MCD) or by estimating a conditional distribution first and then sampling from it (all MDNs).

the test period is from 01 October 1995 to 01 September 2005. This means that we use around $365 * 10 = 3,650$ training points from 531 catchments (equating to a total of $531 * 3650 = 1,938,150$ observations for training).

For all MDNs we introduced an additional hidden layer to provide more flexibility and adapted the network as required (see Apprendix B). We trained all MDNs with the log-likelihood and the MCD as in Kratzert et al. (2020), except that the loss

was the mean-squared error (as proposed by Gal and Ghahramani, 2016). All hyperparameters were selected as to provide the smallest average deviation from the 1:1 line of the probability plot for each model. For GMM this resulted in 10 components and for CMAL in 3 components (Appendix A).

To make the benchmarking procedure work at the most general level, we employed the setup depicted in Figure 7. This allows that each approach, with the ability to generate samples, can be plugged into the framework (as evidenced by the inclusion of

MCD). For each basin and time step the models either predict the streamflow directly (MCD) or provide a distribution over the streamflow (GMM, CMAL, and UMAL). In the latter case, we then sampled from the distribution to get 7500 sample points for each data point. Since the distributions have infinite support sampled values below $0 \, \mathrm{m^3 \, d^{-1}}$ are possible. In this case, we truncated the distribution by setting the sample to zero. All in all, this resulted in $531 * 3650 * 7500$ simulation points for each model and metric. Exaggerating a little bit we could say that we actually deal with *"multi-point"* predictions here.





**Table 2.** Overview of the different single-point prediction performance metrics. The table is adapted from Kratzert et al. (2020).

| single-point metric | Description | Reference |
|---|---|---|
| NSE | Nash–Sutcliffe efficiency | Eq. 3 in Nash and Sutcliffe (1970) |
| KGE | Kling–Gupta efficiency | Eq. 9 in Gupta et al. (2009) |
| Pearson's r | Pearson correlation between observed and simulated flow | |
| $\alpha$-NSE | Ratio of standard deviations of observed and simulated flow | From Eq. 4 in Gupta et al. (2009) |
| $\beta$-NSE | Ratio of the means of observed and simulated flow | From Eq. 10 in Gupta et al. (2009) |
| FHV | Top 2% peak flow bias | Eq. A3 in Yilmaz et al. (2008) |
| FLV | Bottom 30% low flow bias | Eq. A4 in Yilmaz et al. (2008) |
| FMS | Bias of the slope of the flow duration curve between the 20% and 80% percentile | Eq. A2 Yilmaz et al. (2008) |
| Peak-Timing | Mean peak time lag (in days) between observed and simulated peaks | Appendix D in Kratzert et al. (2020) |

## 2.4 Post-hoc Model Examination: Checking Model Behavior

We performed a post-hoc model examination as a complement to the benchmarking to avoid potential blind spots. The analysis has of three parts, each one is associated with a specific property:

1. **Accuracy**: How accurate are single-point predictions obtained from the distributional predictions?

2. **Internal consistency**: How are the mixture components used with regard to flow conditions?

3. **Estimation quality**: How can we examine the properties of the distributional predictions with regard to second-order uncertainties?

### 2.4.1 Accuracy: Single-Point Predictions

To address accuracy, we used standard performance metrics applied to single-point predictions (such as the NSE and the KGE Table 2). The term *single-point predictions* is used here in the statistical sense of a point estimator, to distinguish it from distributional predictions. Single-point predictions were derived as the mean of the distributional predictions at each timestep, and evaluated for aggregating over the different basins, using mean and median as aggregation operators (as in Kratzert et al., 2019b).

### 2.4.2 Internal Consistency: Mixture Component Behavior

To get an impression of the model consistency we looked at the behavioral properties of the mixture densities themselves. The goal was to get some qualitative understanding about how the mixture components are used in different situations. As a prototypical example for this kind of examination we refer to the study of Ellefsen et al. (2019). It examined how LSTMs use the mixture weights to predict the future within a simple game setting. Similarly, Nearing et al. (2020a) reported that a GMM





produce probabilities that change in response to different flow regimes. We conducted the same exploratory experiment with the best-performing benchmarked approach.

### 2.4.3 Estimation Quality: Second Order Uncertainty

MDNs allow a quality check of the given distributional predictions. The basic idea here is that predicted distributions are estimations themselves. MDNs provide an estimation of the aleatoric uncertainty in the data and the MCD is a basic estimation of the epistemic uncertainty. Thus, the estimations of the uncertainties are not the uncertainties themselves, but – as the name suggests – estimations thereof. And, thus subject to uncertainties themselves. This does, of course, hold for all forms of uncertainty estimates, not just for MDNs. However, MDNs provide us with single-point predictions of the *distribution parameters and mixture weights*. We can therefore assess the uncertainty of the estimated mixture components, by checking how perturbations (e.g. in form of input noise) influence the distributional predictions. This can be important in practice. For example, if we mistrust a given input – let us say because the event was rarely observed so far or because we suspect some form of errors – we can use a second order check to obtain qualitative understanding of the goodness of the estimate.

Concretely, we examined how a second order effect on the estimated uncertainty can be checked with MCD approach (which provides estimations for some form of epistemic uncertainties), as it can be layered on top of the MDN approaches (which provide estimations of the aleatoric uncertainties). This means that the Gaussian Process interpretation by Gal and Ghahramani (2016) can not be strictly applied. We can nonetheless use the MCD as a perturbation method, since it still forces the model to learn an internal ensemble.

## 3 Results

### 3.1 Benchmarking Results

The probability plots for each model are shown in Figure 8. The approaches that used mixture densities performed better than MCD, and among all of them, the ones that used asymmetric components (CMAL and UMAL) performed better than GMM. CMAL has the best performance overall. All methods, except UMAL, tend to give estimates above the 1:1 line for thresholds lower than 0.5 (the median). This means that the models were generally under-confident in low-flow situations. GMM was the only approach that showed this type of under-confidence throughout all flow regimes – in other words, GMM was above the 1:1 line everywhere. The largest under-confidence occured for MCD in the mid-flow range (between 0.3 and 0.6 quantile thresholds). For higher flow volumes, both UMAL and MCD underestimated the uncertainty. Overall, CMAL was close to the 1:1 line.

Figure 9 shows how the deviations from the 1:1 line varied for each basin within each threshold of the probability plot. That is, each subplot shows a specific threshold, and each density resulted from the distributions of deviations from the 1:1 line that the different basins exhibit. The distributions for 0.4 to 0.6 flow quantiles were roughly the same across methods, however, the distributions from CMAL and UMAL were better centered than GMM and MCD. At the outer bounds, a bias was induced

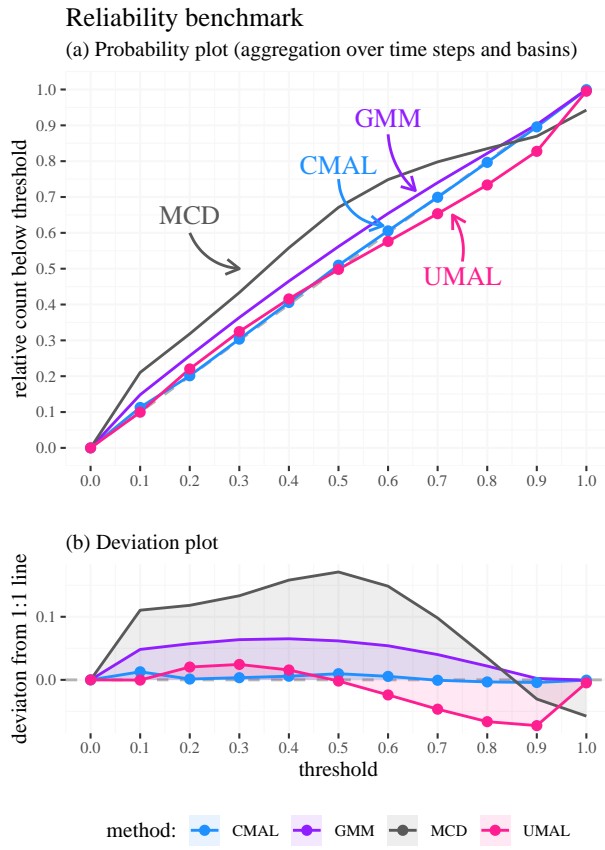

**Figure 8.** Probability plot benchmark results for the 10-year test period over 531 basins in the continental US. Subplot (a) shows the probability plots for the four methods. The 1:1 line is shown in grey and indicates perfect probability estimates. Subplot (b) details deviations from the 1:1 line to allow for easier interpretation.

due to evaluating in probability space: it is more difficult to be over-confident as the thresholds get lower; and vice versa it is more difficult to be under-confident as the thresholds become higher. At higher thresholds, UMAL had a larger tendency to fall below the center-line, which is also visible in the probability plot. Again, this is a consequence of the overconfident predictions from the UMAL approach for larger flow volumes (MCD also exhibited the same pattern of overconfidence for the highest threshold).

Lastly, Table 3 shows the results of the resolution benchmark. In general, UMAL and MCD provide the sharpest distributions. This goes along with overconfident narrow distributions that both approaches exhibit for high flow volumes. These, having the largest uncertainties, also influence the average resolution the most. The other two approaches, GMM and CMAL, provide lower resolution (less sharp distributions). In the case of GMM, the low resolution is reflected in under-confidently wide distributions in the probability plot. Notably, the predictions of CMAL are in between those of the over-confident UMAL and



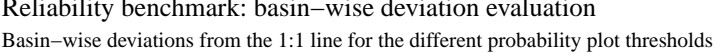

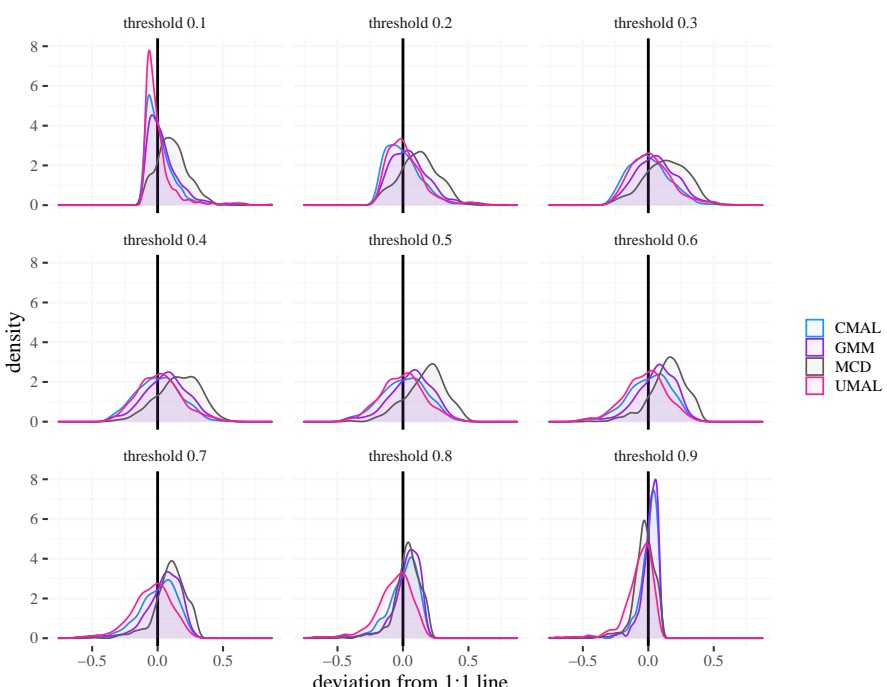

**Figure 9.** Kernel densities of the basin-wise deviation from the 1:1 line in the probability plot for the different inner quantiles. These distributions result from evaluating the performance at each basin individually (rather than aggregating over basins). Note how the bounded domain of the probability plot induces a bias for the outer thresholds as the deviations cannot expand beyond the [0,1] interval.

the under-confident GMM. This makes sense from a methodological viewpoint since we designed CMAL as an *"intermediate*

*step"* between the two approaches. Moreover, these results reflect a trade-off in the log-likelihood (which the models are trained for), where a balance between reliability and resolution has to be obtained in order to minimize the loss.

## 3.2 Post-hoc Model Examination

### 3.2.1 Accuracy

Table 4 shows accuracy metrics of single-point predictions, i.e., the means of the distributional predictions aggregated over

time steps and basins. It depicts the means and medians of each metric across all 531 basins. The approach labeled *MCDp* reports the statistics from the MCD model, but without sampling (i.e., from the full model). The model performances are not entirely comparable to each other, since the architecture and hyperparameters of the MCD model were chosen with regard to the probability plot. We therefore also compare against a model with the same hyper-parameters as Kratzert et al. (2019b) – the latter model is labeled *LSTMp* in Table 4.





**Table 3.** Benchmark statistics for model precision. These metrics were applied to the distributional predictions at individual time steps. The lowest metric per row is marked in bold. Lower values are better for all statistics (conditional on the model having high reliability). This table also provides statistics of the empirical distribution from the observations (obs) aggregated over the basins as a reference, which are not directly comparable with the model statistics since obs represents an unconditional density, while the models provide a conditional one. The obs statistics should be used as a reference to contextualize the statistics from the modeled distributions.

| Benchmarking Metric [a] | GMM | CMAL | UMAL | MCD | obs |
|---|---|---|---|---|---|
| Mean absolute deviation | 0.52 | 0.48 | 0.42 | **0.39** | *0.77* |
| Standard deviation | 0.69 | 0.63 | **0.00**[b] | 0.38 | *2.85* |
| Variance | 2.73 | 2.64 | **0.00**[b] | 0.48 | *12.78* |
| Average with of the 0.2 to 0.9 quantiles | 0.18 | 0.17 | 0.14 | **0.13** | *0.41* |
| Distance between the 0.25 and 0.75 quantiles | 0.71 | 0.68 | 0.67 | **0.51** | *1.38* |
| Distance between 0.1 and 0.9 quantiles | 2.00 | 1.90 | 1.72 | **1.26** | *5.32* |

[a] *All metrics are computed for the samples of each timestep and then averaged over time and basins. All metrics are defined in the domain $[0, \infty)$ and values close to zero are best (conditional on the reliability).*

[b] *The displayed 0 is a rounding artifact. The actual variance here is higher than 0. The "collapse" is, by and large, a result of a very narrow distribution, combined with a heavy truncation for values below zero.*

Among the uncertainty estimation approaches, the models with asymmetric mixture components (CMAL and UMAL) perform best. UMAL provided the best point estimates. This is in line with the high resolutions of the uncertainty estimation benchmark: the sharpness makes the mean a better predictor of the likelihood's maximum and indicates again that the approach trades reliability for accuracy. That said, even with our naive approach for obtaining single-point estimations (i.e., simply taking the mean), both CMAL and UMAL manage to outperform the model that is optimized for single-point pre-
dictions with regard to some metrics. This suggests that it could make sense to train a model to estimate distributions and then recover the best estimates. One possible reason why this might be the case is that single-point loss functions (e.g., MSE) define an implicit probability distribution (e.g., minimizing an MSE loss is equivalent to maximizing a Gaussian likelihood with fixed variance). Hence, using a more nuanced loss function (i.e., one that is the likelihood of a multimodal, asymmetrical, heterogeneous distribution) can improve performance even for the purpose of making non-distributional estimates. In fact, it
is reasonable to expect that the results of the MDN approaches can be improved even further by using a more sophisticated strategy for obtaining single-point predictions (e.g., searching for the maximum of the likelihood). The single-point prediction LSTM (*LSTMp*) outperforms the ALD-based MDNs for tail metrics of the streamflow – that is, for the low- (FLV) and high-





**Table 4.** Evaluation of the different single-point prediction performance metrics. Best performance is marked in bold.

|  | Aggregation | GMM | CMAL | UMAL | MCD | MCDp | LSTMp |
|---|---|---|---|---|---|---|---|
| NSE[a] | median | 0.744 | 0.784 | **0.791** | 0.762 | 0.763 | 0.762 |
| NSE[a] | mean | 0.690 | 0.735 | **0.749** | 0.646 | 0.675 | 0.683 |
| KGE[b] | median | 0.728 | 0.748 | 0.785 | 0.730 | 0.737 | **0.791** |
| KGE[b] | mean | 0.685 | 0.714 | **0.745** | 0.525 | 0.622 | 0.710 |
| COR[c] | median | 0.880 | 0.901 | **0.903** | 0.890 | 0.890 | 0.891 |
| COR[c] | mean | 0.857 | 0.876 | **0.880** | 0.866 | 0.865 | 0.871 |
| $\alpha$-NSE[d] | median | 0.816 | 0.820 | 0.863 | 0.877 | 0.880 | **0.952** |
| $\alpha$-NSE[d] | mean | 0.822 | 0.828 | 0.858 | 0.893 | 0.900 | **0.976** |
| $\beta$-NSE[e] | median | **0.006** | -0.013 | -0.027 | 0.061 | 0.054 | 0.027 |
| $\beta$-NSE[e] | mean | **0.004** | -0.011 | -0.030 | 0.095 | 0.065 | 0.011 |
| FHV[g] | median | -17.322 | -17.164 | 12.243 | -11.346 | -11.343 | **-4.277** |
| FHV[g] | mean | -16.324 | -15.140 | -12.705 | -8.641 | -8.744 | **-1.084** |
| FLV[e] | median | 28.561 | 28.442 | 27.954 | 43.830 | -65.762 | **-4.864** |
| FMS[h] | median | -7.346 | -5.443 | **-2.508** | -20.768 | -17.039 | -8.650 |
| Peak-Timing[i] | median | 0.308 | 0.333 | **0.286** | **0.286** | **0.286** | **0.286** |
| Peak-Timing[i] | mean | 0.464 | 0.455 | 0.412 | 0.427 | 0.425 | **0.405** |

[a] *Nash–Sutcliffe efficiency* $(-\infty, 1]$, *values closer to one are desirable.*

[b] *Kling–Gupta efficiency* $(-\infty, 1]$, *values closer to one are desirable.*

[c] *Pearson correlation:* $[-1, 1]$, *values closer to one are desirable.*

[d] $\alpha$*-NSE decomposition:* $(0, \infty)$, *values close to one are desirable.*

[e] $\beta$*-NSE decomposition:* $(-\infty, \infty)$, *values close to zero are desirable.*

[f] *Top 2 % peak flow bias:* $(-\infty, \infty)$, *values close to zero are desirable.*

[g] *30 % low flow bias:* $(-\infty, \infty)$, *values close to zero are desirable. Since a strong bias is induced by a small subset of basins, we provide the median aggregation.*

[h] *Bias of FDC mid segment slope:* $(-\infty, \infty)$, *values close to zero are desirable. Since a strong bias is induced by a small subset of basins, we provide the median aggregation.*

[i] *Lag of peak timing:* $(-\infty, \infty)$, *values close to zero are desirable.*

bias (FHV). These are regimes where we would expect the most asymmetric distributions for hydrological reasons, and hence the means of the asymmetric distributions might be a suboptimal choice.

### 3.2.2 Internal Consistency

Figure 10 summarizes the behavioral patterns of the CMAL mixture components. It depicts an exemplary hydrograph superimposed on the CMAL uncertainty prediction, together with the corresponding mixture component weights. The mixture weights



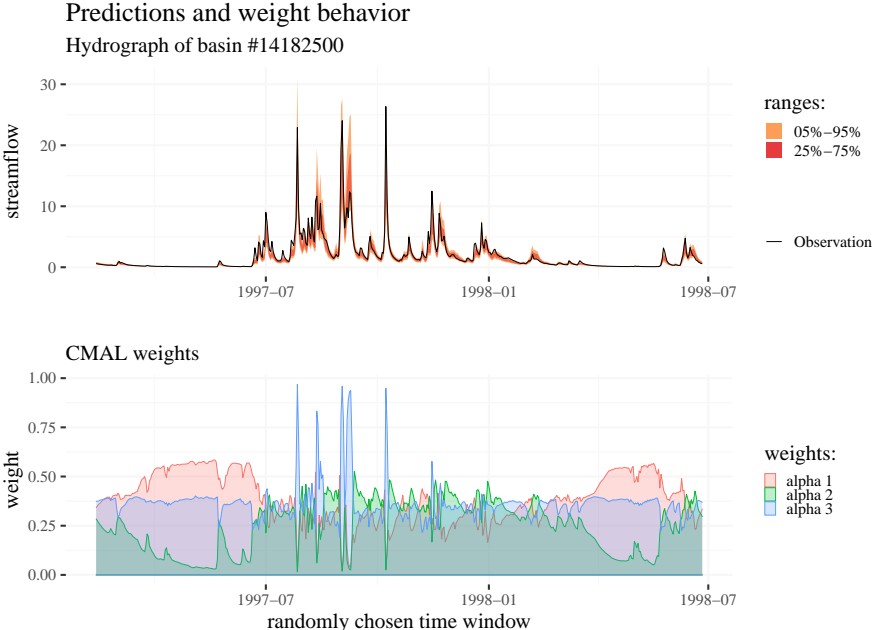

**Figure 10.** Top: Hydrograph of an exemplary event in the test period with both 5% to 95% and 25% to 75% quantile-range. Bottom: The weights ($\alpha_i$) of the CMAL mixture components for these predictions.

always sum to one. This figure shows that the model seemingly learns to use different mixture components for different parts of the hydrograph. In particular, distributional predictions in low-flow periods (perhaps dominated by base flow) are largely

controlled by the first mixture component (as can be seen by the behavior of mixture $\alpha_1$ in Figure 10). Falling limbs of the hydrograph (corresponding roughly to throughflow) are associated with the second mixture component ($\alpha_2$), which is low for both rising limbs and low-flow periods. The third component ($\alpha_3$) mainly controls the rising limbs, the peak-runoff, but also has some influence throughout the rest of the hydrograph. In effect, CMAL learns to separate the hydrograph into three parts – rising limb, falling limb, and low-flows – which correspond to the standard hydrological conceptualization. No explicit

knowledge of this particular hydrological conceptualization is provided to the model – it is solely trained to maximize overall likelihood.

### 3.2.3  Estimation Quality

In this experiment, we are interested in how uncertainties in the estimation of weights and parameters of the mixtures influence the distributional predictions. Figure 11 illustrates the procedure: The upper part shows a hydrograph with the 25%–75%

quantiles and 5%–95% quantiles from CMAL. This is the main prediction. The lower plots show kernel density estimates for particular points of the hydrograph (marked in the upper part with black ovals labeled 'a', 'b' and 'c', and shown in red in the lower subplots). These three specific points represent different portions of the hydrograph with different predicted

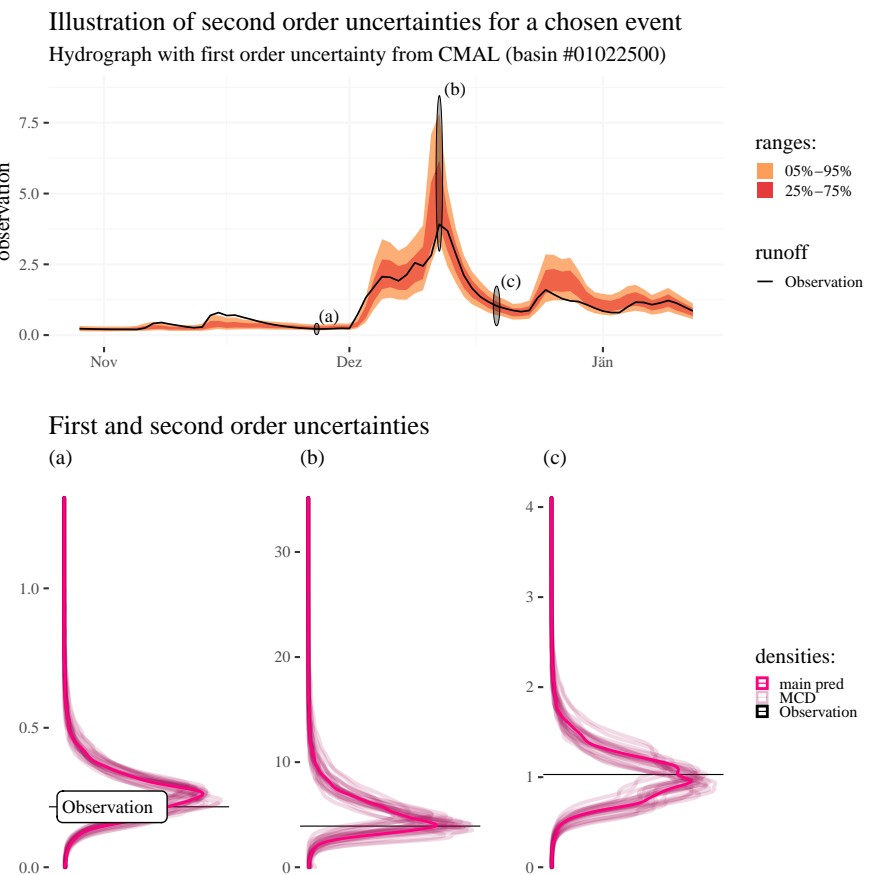

**Figure 11.** Illustration of second order uncertainties estimated by using MCD to sample the parameters of the CMAL approach. The upper subplot shows an observed hydrograph and predictive distributions as estimated by CMAL. The lower subplots show the CMAL distributions and distributions from twenty-five MCD samples of the CMAL model at three selected time steps (indicated by black ovals shown on the hydrograph). The abbreviation *"main pred"* marks the unperturbed distributional predictions from the CMAL model.

distributional shapes and are thus well suited for showcasing the technique. These kernel densities (in red) are superimposed with 25 sampled estimations derived after applying MCD on top of the CMAL model (shown in lighter tones behind the

first order estimate). These densities are the MCD-perturbed estimations and thus a gauge for how second order uncertainties influence the distributional predictions.

## 3.3 Computational Demand

This section gives an overview of the computational demand required to compute the different uncertainty estimations. All of the reported execution times were otbained by using NVIDIA P100 (16GB RAM), using the Pytorch library (Paszke et al.,

2019). A single execution of the CMAL model with a batch size of 256 takes $0.026^{+0.001}_{-0.002}$ seconds (here, and in the following,





the basis gives the median over 100 model runs; the index and the exponent show the deviations of the 10% quantile and the 90% quantile, respectively). An execution of the MCD model takes $0.055^{+0.002}_{-0.001}$ seconds. The slower execution time of the MCD approach here is explained by its larger hidden size. It used 500 hidden cells, in comparison to the 250 hidden cells of CMAL (see Appendix A).

Generating all the needed samples for the evaluation with MCD and a batch size of 256 would take approximately 360 days (since 7500 samples have to be generated for 531 basins and 10 years on a daily resolution). In practice, we could shorten this time to under a week by using considerably larger batch-sizes and distributing the computations for different basins over multiple GPUs. In comparison, computing the same amount of samples by re-executing the CMAL model would take around 174 days. In practice, however, only a single run of the CMAL model is needed, since MDNs provide us with a density estimate

from which we can directly sample in a parallel fashion (and without needing to re-execute the model run). Thus, the CMAL model, with a batch size of 256, takes only ~14 hours to generate all needed samples.

## 4    Conclusions and Outlook

Benchmarking hydrological models is a large undertaking because setting up a model over many watersheds requires substantial effort. Benchmarking uncertainty estimation approaches requires even more effort: In addition to setting up at least

one hydrological model, most uncertainty estimation strategies require the setup and calibration of additional parts to predict uncertainty. These additions might be sampling distributions, ensembles, post-processors, and so forth. Regardless of these difficulties, uncertainty estimation is critical for hydrological forecasting. However, as of now, there is no way to assess different uncertainty estimation strategies for general or particular setups. One of our goals was thus to provide a public data-driven benchmarking scheme. Based on this scheme, other modeling groups can establish a more formal, or at least community-

minded, benchmarking. Our hope is that this will encourage good practice and provide a foundation for others to build on.

With respect to the uncertainty estimation baselines presented here, results using Deep Learning seem promising.The distributional predictions of the uncertainty estimation approaches tested here are dynamic in that the probability distributions change in response to dynamic inputs (precipitation, etc.). Their general behavior is in accordance with basic hydrological intuition in that the uncertainty increases with a rise in streamflow. This relationship is, however, non-linear and not a simple

1:1 depiction. The MCD approach provided the worst uncertainty estimates. One reason for this is likely the Gaussian assumption of the uncertainty estimates, which seems inadequate for many low- and high-flow situations. The (conditional) data is effectively distributed differently. There is however also a more nuanced aspect to consider: The MDN appraoches estimate the aleatoric uncertainty. MCD, on the other hand, estimates epistemic uncertainty, or rather a particular form thereof. The methodological comparison is therefore only partially fair. In general these two uncertainty types can be seen as perpendicular

to each other. They do partially co-appear in our setup, since both the epistemic and the aleatoric uncertainties are largest for high flow volumes.

There was an advantage to using asymmetric distributions as exhibited by UMAL and CMAL, as compared to GMM both in terms of reliability of distributional predictions and accuracy of derived single-point predictions. The CMAL approach in




particular gave distributional predictions that were very good in terms of reliability and sharpness (and single-point estimates).
We could demonstrate that it exhibits a direct link between the predicted probabilities and hydrologic behavior in that different
distributions were activated (i.e., got larger mixture weights) for rising vs. falling limbs, etc.

Nevertheless, likelihood based approaches (for estimating the aleatoric uncertainty) are prone to give over-confident pre-
dictions. We were not able to diagnose this empirically. This might be more a result of the limits of the inquiry than the
non-existence of the phenomenon. Thus, to conclude, we will discuss the limits of the chosen benchmarking setup and the
post-hoc examination:

Our personal observation regarding *benchmark data* is that the current generation of open datasets is the result of a growing
enthusiasm for community efforts. New benchmarking possibilities will become available in the near future. However, a down-
side of relying on open datasets is missing defense against over-fitting on the test data. We therefore hope that in the future the
development of datasets will withhold some test data to make even more rigorous benchmarking possible.

With regard to the *metrics*, the proposed methods exhaust the diagnostic capacity of the probability plot (at least as long
as we use it in the proposed form). This is an encouraging sign for our ability to make reliable hydrologic predictions. The
downside is that it might be hard for models to improve on this metric going forward. As mentioned throughout the manuscript,
it is important to be aware that the probability plot misses several important aspects of probabilistic prediction (for example,
precision, consistency, or event specific properties). We thus expect that future efforts will derive and use more powerful
metrics, similar to the way we continue to develop diagnostic tooling for single-point predictions (see: Nearing et al., 2018).

Stronger *baselines* will then emerge in tandem with stronger metrics. From our perspective, there is plenty of room to build
better ones. A perhaps self-evident example for the potential of improvements are ensembles: Kratzert et al. (2019b) showed
the benefit of LSTM ensembles for single-point predictions, and we believe that similar approaches could be developed for
uncertainty estimation. We are therefore sure that future research will yield improved approaches, and move us closer to
achieving holistic diagnostic tools for the evaluation of uncertainty estimations (*sensu* Nearing and Gupta, 2015).

Our *post-hoc examination* demonstrated how the uncertainty estimation models can be analyzed with regard to hydrological
points of interest. Remarkably, one of the results showed that the distributional predictions are not only reliable and precise
but also yield strong single-point estimates. By and large, however, this just a start. Rainfall–runoff modeling is a complex
endeavor. Often we do not just care that models perform well, but also about the way these predictions are made. Ultimately,
we want to get the *"right answers for the right reasons"* (Kirchner, 2006). Current data is noisy, and metrics do not capture
everything we want models to achieve (see also: Muller, 2018; Thomas and Uminsky, 2020). We therefore expect that, at least
as of now, there will remain blind spots in comparative studies. Post-hoc examination can potentially remedy this fact and
demonstrate to users how model checking can be done. We argue that post-hoc examination will play a central role in future
benchmarking efforts and model developments.



**Table A1.** Overview of the general benchmarking setup.

|  | GMM | CMAL | UMAL | MCD |
|---|---|---|---|---|
| Training Period | 01Oct1980 - 30Sep1990 | 01Oct1980 - 30Sep1990 | 01Oct1980 - 30Sep1990 | 01Oct1980 - 30Sep1990 |
| Validation Period | 01Oct1990 - 30Sep1995 | 01Oct1990 - 30Sep1995 | 01Oct1990 - 30Sep1995 | 01Oct1990 - 30Sep1995 |
| Test Period | 01Oct1995 - 30Sep2005 | 01Oct1995 - 30Sep2005 | 01Oct1995 - 30Sep2005 | 01Oct1995 - 30Sep2005 |
| Training Loss | Negative Log-likelihood | Negative Log-likelihood | Negative Log-likelihood | MSE |
| Camels Attributes | Yes | Yes | Yes | Yes |
| Input Products | DayMet, Maurer, NLDAS | DayMet, Maurer, NLDAS | DayMet, Maurer, NLDAS | DayMet, Maurer, NLDAS |
| Regularization: Noise | Yes | Yes | Yes | Yes |
| Regularization: Dropout | Yes | Yes | Yes | Yes |
| Sampling space for $\tau$ | N/A | N/A | N/A | (0.01,0.99) |
| Gradient clipping | Yes | Yes | No | Yes |

*Code and data availability.* We will make the code for the experiments and data of all produced results available online. We trained all our machine learning models with the `neuralhydrology` Python library (https://github.com/neuralhydrology/neuralhydrology). The CAMELS dataset with static basin attributes is accessible at https://ral.ucar.edu/solutions/products/camels.

## Appendix A: Hyperparameter Search and Trainig

### A1 General Setup

Table A provides the general setup for the hyperparameter search and model training.

### A2 Noise Regularization

Adding noise to the data during training can be viewed as a form of data augmentation and regularization that biases towards smooth functions. These are large topics in themselves, and at this stage we refer to Rothfuss et al. (2019) for an investigation on the theoretical properties of noise regularization and some empirical demonstrations. In short, plain maximum likelihood

estimation can lead to strong over-fitting (resulting in a spiky distribution that generalizes poorly beyond the training data). Training with noise regularization results in smoother density estimates that are closer to the true conditional density.

Following these findings we also add noise as a smoothness regularization for our experiments. Concretely, we decided to use a relative additive noise as a first order approximation to the sort of noise contamination we expect in hydrological time series. The operation for regularization is:

$$\boldsymbol{z}_* = \boldsymbol{z} + \boldsymbol{z} \cdot \mathcal{N}(0, \sigma),$$





**Table A2.** Search space of the hyperparameter search. The search is conducted in two steps: The variables used in first step are shown top part of the table, the variables used in the second step bottom part and written in gray.

|  | GMM | CMAL | UMAL | MCD |
|---|---|---|---|---|
| Hidden size LSTM | 250,500,750,1000 | 250,500,750,1000 | 250,500,750,1000 | 250,500,750,1000 |
| Number of components | 1,3,5,10 | 1,3,5,10 | N/A | N/A |
| Regularization: Noise | 0.05,0.1,0.2 | 0.05,0.1,0.2 | 0.05,0.1,0.2 | 0.05,0.1,0.2 |
| Regularization: Dropout | 0.4,0.5 | 0.4,0.5 | 0.4,0.5 | 0.1,0.25,0.4,0.5,0.75 |
| Batch size | 128,256 | 128,256 | 128,256 | 128,256 |
| Learning rate | 0.0001,0.0005,0.001 | 0.0001,0.0005,0.001 | 0.0001,0.0005,0.001 | 0.0001,0.0005,0.001 |

where $z$ is a placeholder variable for either the dynamic or static input variables or the observed runoff (and, as before, the time index is omitted for the sake of simplicity), $\mathcal{N}(0, \sigma)$ denotes a Gaussian noise term with mean zero and a standard deviation $\sigma$, and $z_*$ the obtained noise contaminated variable.

## A3 Search

To provide a meaningful comparison we conducted a hyperparameter search for each of the four conditional density estimators.
A hyperparameter search is an extended search (usually computationally intensive) for the best pre-configuration of a machine learning model.

In our case we searched over the combination of six different hyperparameters (see Table A2). To balance between computational resources and search depth we took the following course of action:

– First, we informally searched for sensible general presets.

– Second, we trained the models for each combinations of the four hyperparameters "Hidden Size" (the number of cells in the LSTM, see Kratzert et al., 2019b), "Noise" (added relative noise of the output, see Appendix A2), "number of densities" (the number of density heads in the mixture (only needed of MDN and CMAL), and "Dropout Rate" (the rate of dropout employed during training (and inference in the case of MCD)). We marked these in Table A2 with a white background.

– Third, we choose the best resulting model and refine the found models by searching for the best settings for the hyperparameters "batch size" (the number of samples shown per backpropagation step) and "learning rate" (the parameter for the update per batch).

## A4 Results

The results of the hyperparameter search are summarized in Table A3.





**Table A3.** Resulting parameterization from the hyperparaemter search.

|                         | GMM   | CMAL   | UMAL   | MCD   |
|-------------------------|-------|--------|--------|-------|
| Hidden size LSTM        | 250   | 250    | 250,   | 500   |
| Number of components    | 10    | 3      | N/A    | N/A   |
| Regularization: Noise   | 0.2   | 0.2    | 0.2    | 0.1   |
| Regularization: Dropout | 0.4   | 0.5    | 0.5    | 0.75  |
| Batch size              | 256   | 256    | 256    | 256   |
| Learning rate           | 0.001 | 0.0005 | 0.0005 | 0.001 |





## Appendix B: Baselines

### B1 Gausian Mixture Model

Gaussian mixture models (GMMs; Bishop, 1994) are well-established for producing distributional predictions from a single single input. The principle of GMMs is to have a nerual network that predicts the parameters of a mixture of Gaussians (i.e., the means, standard deviations and weights) and to use these mixtures as distributional output. GMMs are a powerful concept. The seen usage for diverse applications such as acoustics (Richmond et al., 2003), handwriting generation (Graves, 2013), sketch generation (Ha and Eck, 2017), and predictive control (Ha and Schmidhuber, 2018).

Given the rainfall-runoff modelling context, a GMM models the runoff $q \in \mathbb{R}$ at a given time step (subscript omitted for the sake of simplicity) as a probability distribution $p(\cdot)$ of the input $\mathbf{x} \in \mathbb{R}^{M \times T}$ (where $M$ indicates the number of defined inputs, such as precipitation and temperature; and $T$ the number of used time steps which are provided to the Neural Network) as a mixture of $K$ Gaussians:

$$p(q \mid \mathbf{x}) = \sum_{k=1}^{K} \alpha_k(\mathbf{x}) \cdot \mathcal{N}\Big(q \mid \mu_k(\mathbf{x}), \sigma(\mathbf{x})\Big), \tag{B1}$$

where $\alpha_k$ are a mixture weights with the property $\alpha_k(\mathbf{x}) \geq 0$ and $\sum_{k=1}^{K} \alpha_k(\mathbf{x}) = 1$ (convex sum); and $\mathcal{N}(\mu_k(\mathbf{x}), \sigma_k(\mathbf{x}))$ denotes a Gaussian with mean $\mu_k$ and standard deviation $\sigma_k$, All three defining variable – i.e. the mixture weights, the means, and the standard deviations, are set by a Neural Network and thus a function of the inputs $\mathbf{x}$.

The negative logarithm of the likelihood between the training data the training data and the estimated conditional distribution a is used as loss, i.e.:

$$L(q \mid \mathbf{x}) = -\log\left[\sum_{k=1}^{K} \alpha_k(\mathbf{x}) \cdot \mathcal{N}\Big(q \mid \mu_k(\mathbf{x}), \sigma_k(\mathbf{x})\Big)\right]. \tag{B2}$$

For the actual implementation we used a softmax activation function to obtain the mixture weights ($\alpha$); and an exponential function as activation for variance ($\sigma$) to guarantee that the estimate is always above zero (see: Bishop, 1994)

### B2 Countable Mixture of Asymmetric Laplacians

Countable mixtures of asymmetric Laplacian distributions, short CMAL, are another form of MDN where asymmetric Laplacian distributions (ALDs) are used as a kernel function. The acronym is a reference to UMAL since it serves as a natural intermediate intermediate stage between GMM and UMAL – as will become clear in the respective section. As far as we are aware, the use of ALDs for quantile regression was proposed by Yu and Moyeed (2001) and their application for MDNs was first proposed by Brando et al. (2019). The components of CMAL already intrinsically provide a measure for asymmetric



distributions and are therefore inherently more expressive than GMM. However, since they also necessitate the estimation of more parameters one can expect that they are also more difficult to handle than GMMs. The density for the ALD is:

$$
\mathcal{A}_{LD}(q \mid \mu, s, \tau) = \frac{\tau \cdot (1-\tau)}{b} \cdot
\begin{cases}
\exp\left[-(q-\mu) \cdot (\tau-1)/s\right], & \text{if } q < \mu \\
\exp\left[-(q-\mu) \cdot \tau/s\right], & \text{if } q \geq \mu
\end{cases}
\tag{B3}
$$

where $\tau$ is asymmetry parameter, $\mu$ the location parameter and $s$ the scale parameter respectively. Using the ALD as a
component CMAL can be defined in analogy to the GMM:

$$
p(q \mid \mathbf{x}) = \sum_{k=1}^{K} \alpha_k(\mathbf{x}) \cdot \mathcal{A}_{LD}\Big(q \mid \mu_k(\mathbf{x}), s_k(\mathbf{x}), \tau_k(\mathbf{x})\Big),
\tag{B4}
$$

and the parameters and weights are estimated by a Neural Network. Training is done by maximizing the negative log-likelihood of the the training data from estimated distribution:

$$
L(q \mid \mathbf{x}) = -\log\left(\sum_{k=1}^{K} \alpha_k(\mathbf{x}) \cdot \mathcal{A}_{LD}\Big(q \mid \mu_k(\mathbf{x}), s_k(\mathbf{x}), \tau_k(\mathbf{x})\Big)\right).
\tag{B5}
$$

For the implementation i the network we used a softmax activation function to obtain the mixture weights ($\alpha$); a sigmoid
function to bind the asymmetry parameters ($\tau$); and a softplus activation function to guarantee that the scale ($b$) is always above
zero.

**B3   Uncountable Mixture of Asymmetric Laplacians**

Uncountable Mixture of Asymmetric Laplacians (UMAL; Brando et al., 2019) expands upon the CMAL concept by letting the
model implicitly approximate the mixture of ALDs. This is achieved (a) by sampling the asymmetry parameter $\tau$ and providing
it as input to the model and the loss and (b) by fixing the weights with $\alpha_k = 1/K$ and (c) stochastically approximating the
underlying distributions by summing up different realizations. Since the network only has to account the scale and the location
parameter, considerably less parameters have to be estimated than for the GMM or CMAL.

In analogy to the CMAL model equations, these extensions lead to the following equation for the conditional density:

$$
p(q \mid \mathbf{x}) = \frac{1}{K} \sum_{k=1}^{K} \mathcal{A}_{LD}\big(q \mid \mu_k(\mathbf{x}, \tau_k), s_k(\mathbf{x}, \tau_k), \tau_k\big),
\tag{B6}
$$

where the asymmetry parameter $\tau_k$ is randomly sampled $k$ times to provide a Monte Carlo approximation to an implicitly
approximated distribution. After the training modellers can choose from how many discrete samples the learned distribution





is approximated. As with the other mixture density networks the training is done by minimizing the negative log-likelihood of the training data from the estimated distribution:

$$L(q \mid \mathbf{x}) = -\log\left(\sum_{k=1}^{K} \mathcal{A}_{LD}\Big(q \mid \mu_k(\mathbf{x}, \tau_k), s_k(\mathbf{x}, \tau_k), \tau_k\Big)\right) + \log(K). \tag{B7}$$

Implementation wise we obtained the best results from UMAL by binding the scale parameter ($s_k$). We therefore used a weighted sigmoid function as activation.

### B4 Monte Carlo Dropout

Monte Carlo Dropout (MCD; Gal and Ghahramani, 2016) has found widespread and has already been used in a large variety
of applications (e.g., Zhu and Laptev, 2017; Kendall and Gal, 2017; Smith and Gal, 2018). The MCD mechanism can be can be expressed as:

$$p(q \mid \mathbf{x}) = \mathcal{N}\big(q \mid \mu^*(\mathbf{x}), \sigma_k\big) \tag{B8}$$

where $\mu^*(\mathbf{x})$ is the expectation over the sub-networks given the dropout rate $r$, such that:

$$\mu^*(\mathbf{x}) = \mathrm{E}_{\boldsymbol{d} \sim \mathcal{B}(r)}[f(\mathbf{x}, \boldsymbol{d}, \theta)] \approx \frac{1}{M} \sum_{m=1}^{M} f(\mathbf{x}, \boldsymbol{d_m}, \theta), \tag{B9}$$

where $\boldsymbol{d}$ is a dropout mask sampled from a Bernoulli distribution $\mathcal{B}$ with rate $r$, $\boldsymbol{d_m}$ is a particular realization of a dropout mask, $\theta$ are the network weights, and the $f(\cdot)$ is the neural network. Note $f(\mathbf{x}, \boldsymbol{d_m}, \theta)$ is equivalent to a particular sub-network of $f$.

MCD is trained by maximizing the expectancy, that is by minimizing the mean squared error. As such it is quite different from the MDN appraoches. It provides an estimation of the epistemic uncertainty and as such does not supply a heterogeneous,
multi modal estimate (it assumes a Gaussian form). For evaluation studies of MCD in hydrological fields we refer to Fang et al. (2019) who investigated its usage in the context of soil-moisture prediction. We also note that it has been observed that MCD can underestimate the epistemic uncertainty (e.g., Fort et al., 2019).



*Author contributions.* DK, FK, MG, and GN designed all experiments. DK conducted all experiments and the results where analyzed together with the rest of the authors. FK and MG helped with building the modellign pipeline. FK provided the main setup for the "Accuracy"

analysis; AKS and GN for the "Internal Consistency" analysis, and DK for the "Estimation Quality" analysis. GK and JH checked the technical adequacy of the experiments. GN supervised the manuscript from the hydrologic perspective, SH from the machine learning perspective. All authors worked on the manuscript.

*Competing interests.* The authors declare that they have no conflict of interest.

*Acknowledgements.* This research was undertaken thanks in part to funding from the Canada First Research Excellence Fund and the Global

Water Futures Program, and enabled by computational resources provided by Compute Ontario and Compute Canada. The ELLIS Unit Linz, the LIT AI Lab, and the Institute for Machine Learning are supported by the Federal State Upper Austria. We thank the projects AI-MOTION (LIT-2018-6-YOU-212), DeepToxGen (LIT-2017-3-YOU-003), AI-SNN (LIT-2018-6-YOU-214), DeepFlood (LIT-2019-8-YOU-213), Medical Cognitive Computing Center (MC3), PRIMAL (FFG-873979), S3AI (FFG-872172), DL for granular flow (FFG-871302), ELISE (H2020-ICT-2019-3 ID: 951847), AIDD (MSCA-ITN-2020 ID: 956832). Further, we thank Janssen Pharmaceutica, UCB Biopharma

SRL, Merck Healthcare KGaA, Audi.JKU Deep Learning Center, TGW LOGISTICS GROUP GMBH, Silicon Austria Labs (SAL), FILL Gesellschaft mbH, Anyline GmbH, Google (Faculty Research Award), ZF Friedrichshafen AG, Robert Bosch GmbH, Software Competence Center Hagenberg GmbH, TÜV Austria, and the NVIDIA Corporation.



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
