# Peer review of "Uncertainty Estimation with Deep Learning for Rainfall–Runoff Modelling"

_Hydrology and Earth System Sciences, 2021_

## Referee Comment (RC1)

**Review hess-2021-154**

**TITLE**

Uncertainty Estimation with Deep Learning for Rainfall–Runoff Modelling

**RECOMMENDATION**

Moderate Revision

**REVIEWER**

John Quilty

**GENERAL COMMENTS**

This paper introduces several variants of long short-term memory networks (LSTM) for the estimation of predictive uncertainty in rainfall-runoff modelling. The methods are explored using the CAMELS dataset (Catchment Attributes and MEteorology for Large-sample Studies) (Addor et al., 2017) as a means to provide a large-scale benchmark for the proposed methods as well as others that may be explored in the future. My takeaway is that both of these items are the main contributions of the paper, the latter, in my opinion, being the most relevant.

In general, the paper is well-written and clear, the methodology is reasonable (however, some suggestions are included below), and the results seem promising. The main point I feel important to raise is that the authors adopt LSTM as the sole model and do not compare it against a plethora of other 'simpler' data-driven models that can provide estimates of the predictive uncertainty, likely with much less computational cost and onerous hyper-parameter tuning. Since the authors are looking to create a benchmark for comparing different predictive uncertainty estimation techniques in the context of rainfall-runoff modelling, it seems reasonable that very simple approaches be explored as a baseline before considering more complex methods. This critique is not meant to take anything away from the work that is done, since the reviewer understands LSTM may be a preferred method within this group (as is the case with process-based models within other groups), but more to seek justification for using more complicated (computationally intensive) models when it may suffice to use simpler ones that end up providing similar results, which seems relevant to consider from a benchmarking perspective.

Aside form the above point, I consider most other comments below relatively minor but still important to address.

If the authors can suitably address the comments noted below, I would be happy to recommend that the paper be published in Hydrology and Earth System Sciences.

**SPECIFIC COMMENTS**

1. Terminology: throughout the paper the authors mention both 'prediction' and 'forecasting', which have different purposes and (potentially) different modelling setups. The authors should be consistent in their use of terminology throughout the manuscript. It appears the term 'prediction' is more appropriate considering their application.

2. Introduction: in L20-31 it would be good for the authors to acknowledge other established and recent methods for probabilistic prediction (and forecasting) in hydrology including: quantile regression-based neural networks (Cannon, 2018, 2011), copula-based approaches (Li et al., 2021; Liu et al., 2021), and (parametric/distributional) probabilistic decision tree methods (Başağaoğlu et al., 2021; Schlosser et al., 2019), the latter methods allow for the predictive distribution of the target variable to be easily estimated as part of the training procedure. Although I have yet to undertake such an analysis myself, I suspect the latter group of methods could estimate predictive uncertainty with much less computational expense than the LSTM variants adopted here.

3. Introduction: the authors may want to acknowledge the recent paper by Althoff et al. (2021) whom also used MCD with LSTMs for daily streamflow forecasting.

4. Section 2 and 2.3: while I appreciate the authors have been exploring LSTM in many former studies and are evolving a research program around this method, from my own experience, these methods are extremely computationally intensive (due to it's recurrent formulation, gradient-based learning, need for careful hyper-parameter tuning, etc.), and thus tend to be a 'turn-off' for those who do not have adequate computing resources to explore such methods. It would seem beneficial for the reader to have a 'simpler' non-LSTM baseline to compare against the proposed LSTM variants. If the results of the LSTM variants significantly outperform these simpler methods, then it may serve as a motivation for other researchers to devote more time and resources to incorporating LSTM into their research endeavours. My feeling is that a simple benchmark method could include one of the many variants of quantile regression forests (see for example, https://scikit-garden.github.io/examples/QuantileRegressionForests/). This would not seem out of scope as the authors mention in Section 2 that their work is devoted to data-driven methods, of which LSTM are only a small (but relevant) fraction.

5. L177-180: This somewhat confusing. If I understand correctly (based on the Althoff et al. (2021) paper), dropout is used during training at each iteration but it does not create a separate model at each iteration, only a 'thinned' network. However, performing dropout during testing (or model implementation, evaluation, or whatever other term you wish to use), each time you make a prediction you simply turn on/off nodes according to the pre-specified probabilities used during training and you repeat this as many times as you desire, creating a number of 'sister' predictions. Again, the model does not change, you simply 'thin' the network each time you create a 'sister' prediction. If this is how MCD was used in the experiments described in this paper, it is not apparent and would be helpful to clarify.

6. L193-4: why not use a regularized squared loss function? Is it not a standard practice to perform L2 (and potentially L1) regularization to improve LSTM performance and reduce overfitting? Was this considered? If not, why?

7. L195-6: what dataset partition was used to select optimal hyper-parameters?

8. Section 2.4.3: the authors should formally define first and second order uncertainties.

9. Table 3: it's not clear how the 'obs' data is to be used to 'contextualize' the results from the different models. More detail should be provided (perhaps with an example in the relevant

section).

10. L286-289: it would seem like a good idea for the authors to investigate how best to use the CMAL and UMAL variants for improving predictive performance at low and high (tail) flows (from the point prediction point of view), as this tends to be a major impetus for creating models with uncertainty assessment capabilities. In other words, what's the point of going through the trouble of designing these more sophisticated methods if they cannot outperform the base approach (LSTMp) when assessed on highly relevant metrics. Please don't get me wrong, I am not trying to downplay the very interesting work done by the authors, I am simply trying to help them more fully explore the merits of their work and better 'sell' their approach to a 'skeptic'.

11. Section 3.2.2 is very interesting!

12. Section 3.2.3: once first and second order uncertainties are formally defined (see comment 8), this section should give a good description of what is *shown* in Figure 11 but it is unclear what message the authors expect the reader to take-away from this figure. What's the relevance of this figure and why should the reader 'care' about it?

13. Section 3.3: my understanding is that this is the time needed to make predictions with a trained model. What is the training time for the different models? Can the authors provide an example calculation for the overall run-time in Appendix A (or at the very least in their reply, it's not clear how the 365 and 174 days were calculated)?

14. Section 4: after L335 the authors may wish to very briefly summarize the adopted models and datasets used in the study before continuing with L336 onwards. This should help the interested reader with a short 'time-budget', who may only jump from the abstract to the conclusion, get a decent idea of the methods and dataset involved (the dataset being one of the key strengths of this paper).

15. Each equation in the appendices (B1, B2, etc.) should be properly cited (the rule is to cite all equations that are not developed by the authors in the paper).

**TECHNICAL CORRECTIONS**
- L4: 'This contributions…'
- L9: Please rephrase '…and show that learn nuanced behaviors in different situations.'
- L182-3 seems to be repeated at L193-4?
- Table 2 should be placed after it is mentioned in the text.
- L207: remove 'of'.
- Spell out NSE and KGE at first use.
- L216: perhaps indicate where the median aggregator is used in the paper?
- L223: 'produced'.
- L33: 'in **the** form of'.
- Figure 10 and 11 legends: remove the 0 from '05'.
- Figure 11 x-axis: 'Dec' not 'Dez'.

- L417: remove 'single' (duplicated).
- L420-1: 'They have seen...'?
- L430 should be reviewed for duplicated words ('the training data') and typos.
- L450: not clear what 'i' pertains to...
- L470: remove 'can be' (duplicated).

**REFERENCES**

Addor, N., Newman, A.J., Mizukami, N., Clark, M.P., 2017. The CAMELS data set: Catchment attributes and meteorology for large-sample studies. Hydrol. Earth Syst. Sci. 21, 5293–5313. https://doi.org/10.5194/hess-21-5293-2017

Althoff, D., Rodrigues, L.N., Bazame, H.C., 2021. Uncertainty quantification for hydrological models based on neural networks: the dropout ensemble. Stoch. Environ. Res. Risk Assess. 35, 1051–1067. https://doi.org/10.1007/s00477-021-01980-8

Başağaoğlu, H., Chakraborty, D., Winterle, J., 2021. Reliable evapotranspiration predictions with a probabilistic machine learning framework. Water (Switzerland) 13. https://doi.org/10.3390/w13040557

Cannon, A.J., 2018. Non-crossing nonlinear regression quantiles by monotone composite quantile regression neural network, with application to rainfall extremes. Stoch. Environ. Res. Risk Assess. 32, 3207–3225. https://doi.org/10.1007/s00477-018-1573-6

Cannon, A.J., 2011. Quantile regression neural networks: Implementation in R and application to precipitation downscaling. Comput. Geosci. 37, 1277–1284. https://doi.org/10.1016/j.cageo.2010.07.005

Li, H., Huang, G., Li, Y., Sun, J., Gao, P., 2021. A C-Vine Copula-Based Quantile Regression Method for Streamflow Forecasting in Xiangxi River Basin, China. Sustainability 13. https://doi.org/10.3390/su13094627

Liu, Z., Cheng, L., Lin, K., Cai, H., 2021. A hybrid bayesian vine model for water level prediction. Environ. Model. Softw. 142, 105075. https://doi.org/10.1016/j.envsoft.2021.105075

Schlosser, L., Hothorn, T., Stauffer, R., Zeileis, A., 2019. Distributional regression forests for probabilistic precipitation forecasting in complex terrain. Ann. Appl. Stat. 13, 1564–1589. https://doi.org/10.1214/19-AOAS1247

---

## Author Comment (AC1)

**REVIEWER 1: John Quilty**

**GENERAL COMMENTS**

This paper introduces several variants of long short-term memory networks (LSTM) for the estimation of predictive uncertainty in rainfall-runoff modelling. The methods are explored using the CAMELS dataset (Catchment Attributes and MEteorology for Large-sample Studies) ( Addor e t al ., 2017) as a means to provide a large-scale benchmark for the proposed methods as well as others that may be explored in the future. My takeaway is that both of these items are the main contributions of the paper, the latter, in my opinion, being the most relevant. In general, the paper is well-written and clear, the methodology is reasonable (however, some suggestions are included below), and the results seem promising. The main point I feel important to raise is that the authors adopt LSTM as the sole model and do not compare it against a plethora of other 'simpler' data-driven models that can provide estimates of the predictive uncertainty, likely with much less computational cost and onerous hyper-parameter tuning. Since the authors are looking to create a benchmark for comparing different predictive uncertainty estimation techniques in the context of rainfall-runoff modelling, it seems reasonable that very simple approaches be explored as a baseline before considering more complex methods. This critique is not meant to take anything away from the work that is done, since the reviewer understands LSTM may be a preferred method within this group (as is the case with process-based models within other groups), but more to seek justification for using more complicated (computationally intensive) models when it may suffice to use simpler ones that end up providing similar results, which seems relevant to consider from a benchmarking perspective. Aside form the above point, I consider most other comments below relatively minor but still important to address.If the authors can suitably address the comments noted below, I would be happy to recommend that the paper be published in Hydrology and Earth System Sciences.

Thank you for the thoughtful commentary. It was very helpful to us. We will respond to each of these issues in specific comments below, however it is worthwhile to address the larger point up front.

In general, we agree with the sentiment that complex approaches should be compared with a simpler reference. The lack thereof is exactly the reason why we proposed our baselines: We do not see any non-trivial method that would be easier than approaches that only require differentiable model-backbones able to provide the needed outputs. Especially if one considers that we gain fully distributional predictions from them (see Figure I of our answers), which are extremely rich representations that render them very flexible. That is, they can be used for almost all kinds of comparisons and benchmarking efforts in the future (e.g., we can analytically analyse the distribution as such, sample from them, derive point estimates such as the mean, the median or the maximum likelihood estimate, etc.; see also Figure 1 of our ).

Further, just adding arbitrary methods to the paper just because we can do it from a computational standpoint would be antithetical with the idea of letting benchmarking become a community effort: Good benchmarking is not something that can be done in a responsible way in any single contribution, unless that paper itself is the outcome of a larger community effort (e.g., Best et al., 2015; Kratzert et al., 2019). Constructing new benchmarking approaches in a vacuum can be dangerous because they can easily become straw men (i.e., they might appear to be bad in isolation, but in reality the performance just reflects the choices or inabilities of the modellers). It can take a lot of knowledge, skill, and experience in any given method to use it correctly (even for "simple" methods). Because there is so much nuance to the current generation of ML and DL methods, because they take so much knowledge and skill to implement correctly, and because the hydrology community is just starting to build widespread expertise in these areas, it is all too easy to conduct flawed comparative studies in a vacuum. The way to guard against this is community benchmarking: We start with a set of self-contained baselines that are closed in themselves and openly share our framework, data, and methods Over time, we as a

community can then improve, replace, or add to them. Everyone runs the model or approach that they know best and results are compared at a community level.

As an example, in the specific comment 4 of reviewer 1 a particular method (QRF) is suggested to us. This somewhat undermines our intention of establishing DL-based baselines, since it is implied that practitioners can use it as a point of comparison. Imagine if we either naively or nefariously included the results from our QRF exploration (see our answer to specific comment #4) in the paper and showed that we could beat it. The end result would be that we would not have to push back against the request — and also our methods would look better by comparison. Further, the reviewer has not not used the method for the present setting (see specific comment #4), so he would have to trust our evaluation and make our road to publication easier. Despite this, we argue to not include QRF, since including it (and the corresponding outcomes) would be unrigorous . That is, the results might paint a wrong picture of the capabilities of the QRF, thus flawing the comparative benchmark and its potential adaptation.

This is why community benchmarking is so important. Community benchmarking is how a research community guards against this type of ad hoc and potentially misleading comparison. This study provides a UE benchmark on the CAMELS data, which is a large, publicly curated, open dataset that has been used frequently for model benchmarking. Our goal is to start a community benchmarking effort on this dataset for UE. In our view adding ad hoc benchmarks would be counterproductive.

[Figure]

**Figure 1**. Different forms of predictions and their relation to each other. Note that the point prediction in plot (a) is the mean of the distribution prediction of plot (c).

Lastly we would like to comment on the presumption that our use of the LSTM perhaps is a subjective preference: While it might be true that some groups choose which model to use based on subjective preferences instead of objective criteria (e.g., Addor and Melsen, 2018), we do not view our choice as such. We use LSTMs for one reason: they are the best rainfall-runoff model that the hydrological sciences community has so far discovered by almost any metric. As mentioned above, the proposed UE approaches work with any differentiable model able to provide the required outputs. CMAL, UMAL, and GMM are just specific final layers in a DL network. MCD does not even require such a layer. If tomorrow some group developed a model that works better than the LSTM we would switch immediately. And, if said model would be based on the DL approach, say a transformer (Vasvani et al., 2017), the proposed UE baselines could simply be added as a layer on top. The LSTMs here are used only because they are currently the best rainfall-runoff model.

**SPECIFIC COMMENTS**

1. Terminology: throughout the paper the authors mention both 'prediction' and 'forecasting', which have different purposes and (potentially) different modelling setups. The authors should be consistent in their use of terminology throughout the manuscript. It appears the term 'prediction' is more appropriate considering their application.

Thank you for pointing this out. As mentioned in the original manuscript we tried to stick to the convention laid out by Beven and Young (2013) who suggest to use the word 'simulation' for a setting where the model uses a specific input for each time step but does not receive information about the observed outputs, and to use the term 'forecast' for settings where all information up to a given point in time is used to make a prediction. The term 'prediction' can be used for either situation. Thus, we should almost always use the term prediction or simulation here.

We used the term 'forecast' two times incorrectly in the paper - once in the abstract L.2 and once in the conclusions L.332. We will replace these occurrences with the term 'prediction' in the revised manuscript as proposed by the reviewer. This is unfortunate because we explicitly stated (and cited) our convention for this terminology in L.182 to L.186 of the original manuscript, and we appreciate the careful attention by the reviewer in catching this mistake.

2. Introduction: in L20-31 it would be good for the authors to acknowledge other established and recent methods for probabilistic prediction (and forecasting) in hydrology including: quantile regression-based neural networks (Cannon, 2018, 2011), copula-based approaches (Li et al., 2021; Liu et al., 2021), and (parametric/distributional) probabilistic decision tree methods (Başağaoğlu et al., 2021; Schlosser et al., 2019), the latter methods allow for the predictive distribution of the target variable to be easily estimated as part of the training procedure. Although I have yet to undertake such an analysis myself, I suspect the latter group of methods could estimate predictive uncertainty with much less computational expense than the LSTM variants adopted here.

This paper is not intended as a literature review of probabilistic prediction in hydrology in general. There are dozens of methods that the review could have listed here as examples of probabilistic prediction. The original manuscript already covers many established and recent methods that are directly relevant to our study. There are enough "probabilistic prediction" methods in the hydrology literature that if someone wanted to do a comprehensive review, this would end up being a stand-alone review paper. We do not see anything particularly special or relevant about the suggested references. If there is a particular reason to cite a paper that is directly relevant to this study, then we definitely want to cite it — for example, the paper suggested in the next comment.

Regarding the performance: In our eyes, LSTMs are computationally cheap to train and run. As of now, they can be trained and run using large data sets on any modern computer with a GPU in a few hours. For comparison, take the QRF proposed in specific comment 4. It required more resources (CPU) to run at a similar scale (details provided in our answer to specific comment 4).

3. Introduction: the authors may want to acknowledge the recent paper by Althoff et al. (2021) whom also used MCD with LSTMs for daily streamflow forecasting.

Thank you for this reference. It is indeed very relevant and we were not aware of it (as can be seen from the arxiv submissions, we finished our paper before Althof et al. (2021) was published and did not see it in the aftermath). We will include it in the revised manuscript.

4. Section 2 and 2.3: while I appreciate the authors have been exploring LSTM in many former studies and are evolving a research program around this method, from my own experience, these methods are extremely computationally intensive (due to it's recurrent formulation, gradient-based learning, need for careful hyper-parameter tuning, etc.), and thus tend to be a 'turn-off' for those who do not have adequate computing resources to explore such methods. It would seem beneficial for the reader to have a 'simpler' non-LSTM baseline to compare against the proposed LSTM variants. If the results of the LSTM variants significantly outperform these

simpler methods, then it may serve as a motivation for other researchers to devote more time and resources to incorporating LSTM into their research endeavours. My feeling is that a simple benchmark method could include one of the many variants of quantile regression forests (see for example, https://scikit-garden.github.io/examples/QuantileRegressionForests/). This would not seem out of scope as the authors mention in Section 2 that their work is devoted to data-driven methods, of which LSTM are only a small (but relevant) fraction.

The answer to this comment is closely linked to our answer to the general comments of reviewer 1. While the answer there gives our arguments with regard to our vision (and is therefore more important), this answer can be seen as a technical addendum.

First, we would like to emphasize that we use the LSTM because it is the best model that we (or any other hydrology group) has found for rainfall-runoff estimation. This is the only reason. We care about providing information to research and operational groups who want to do the best job possible. Within this "big-data regime" our team has examined many different types of ML and DL models. For example, we have tried transformers, MLPs, boosted regressions (XGBoost), experimented with neural ODEs, and developed custom physics-informed neural network architectures (Hoedt et al., 2020). We will continue to try other approaches whenever possible. So far LSTMs simply work the best for the task of rainfall-runoff modeling.

Second, the LSTM does not constitute an uncertainty estimation approach, as such. In this paper it is used as a base model and the uncertainty approaches are independent of the LSTM in the sense that they could be applied to any type of ML model (they are just a layer in the deep learning network). If tomorrow we discovered that — for example — transformers were better than LSTMs for rainfall-runoff modeling, we would simply apply the UMAL/CMAL/GMM methods tested here to transformers.

Third, and irrespective of all of what we said above, we tested the quantile regression forest (QRF) that the reviewer suggested: In summary, the QRF performed worse with respect to the particular modeling problem. We did not anticipate these problems before starting with QRFs — it seems like a reasonable method — but after

running the method and understanding why it fails, it seems to us that it is not fit for this purpose. We have three things to say about this:

1) QRF doesn't work for regional streamflow modeling. QRF gives a median NSE of ~0.24 (Figure 2), whereas the lowest median NSE value in our work is from GMM at ~0.74. Because the QRF loss function is based on counting bins (it is the MSE of the predicted vs. actual number of observations that fall in each quantile bin), it can learn a perfect probability plot (Figure 3) without learning any dynamics of the system.  In our results it learns some dynamics, but not enough to simulate realistic (probabilistic) hydrographs (Figure 4). The result is that while the average QQ plot looks good (because it is  directly optimized for this metric), it has large biases from basin to basin (Figure 5). QRF is simply not designed for this type of modeling problem (Meinshausen, 2006).

2) The proposed QRF is computationally expensive (much more expensive than the LSTM). Used naively, it would take a long time to train the QRF on the same data for which we can train an LSTM in under 3 hours on a laptop with a single GPU, even when QRF fitting is fully parallelized on a node with 40 cores. It is possible that the specific implementation that the reviewer linked is poorly coded (there might be better ways to implement the QRF algorithm), but we are not experts in this method and we had to make several adaptations (in terms of hyperparameters) to get a performant variant to work.

3) This leads us to our third point, which is that using the QRF is not simple. There are many hyperparameters that must be set based on a combination of intuition, expert knowledge, and formal hyperparameter searching. Informative hyperparameter tuning would require millions of CPU-hours for QRF (as opposed to only hundreds of GPU hours for LSTM hypertuning). Additionally, we are not experts in QRF, so if we were to include any QRF results in this or any other paper, someone who is an expert could likely find criticism of our implementation. This is why community benchmarking is so important - groups who are actually invested in a method need to implement that method in a reproducible way on open community datasets so that results are directly comparable.

[Figure]

**Figure 2**. Empirical cumulative distribution function of the Nash-Sutcliffe Efficiencies (NSE) obtained by using the median of the Quantile Random Forest as a predictor. The x-axis is limited to -1 to 1, the mean NSE is depicted with a red vertical line, and the median NSE with an orange vertical line.

[Figure]

**Figure 3**. Reliability benchmark for the Quantile Random Forest (qrf) and the Countable Mixture of Asymmetric Laplacians (CMAL) in form of the probability plot and the deviation plot in the style of Figure 8 of the original manuscript.

[Figure]

**Figure 4**. Working example hydrograph for a random basin. The observed streamflow is shown in blue, the median in red and the interquartile-range (the distance between the 25th and 75th percentiles) in orange.

[Figure]

**Figure 5**. Demonstration of using the absolute deviations from the 1:1 line as an ad-hoc diagnostic, which does not allow trading-off performances between basins.

5. L177-180: This is somewhat confusing. If I understand correctly (based on the Althoff et al. (2021) paper), dropout is used during training at each iteration but it does not create a separate model at each iteration, only a 'thinned' network. However, performing dropout during testing (or model implementation, evaluation, or whatever other term you wish to use), each time you make a prediction you simply turn on/off nodes according to the pre-specified probabilities used during training and you repeat this as many times as you desire, creating a number of 'sister' predictions. Again, the model does not change, you simply 'thin' the network each time you create a 'sister' prediction. If this is how MCD was used in the experiments described in this paper, it is not apparent and would be helpful to clarify.

We are not sure if we can follow why the reviewer finds these lines confusing.

The reviewer's interpretation of the dropout mechanism is correct, however this is exactly what is described in lines 177-180. The reviewer also correctly assesses that a thinned network is not a separate model in the sense that all thinned networks use the same tensor network. Thus, the reviewer might be objecting to our use of the term "sub-model" in this sentence, however this is the term used in the original dropout paper (Srivastava et al., 2014) to describe the concept: *"The central idea of dropout is to take a large model that overfits easily and repeatedly sample and train smaller sub-models from it."* (Srivastava et al., 2014). In our eyes it is very intuitive to think of dropout as a way of building up an implicit ensemble (as a matter of fact, a very particular one as shown in Gal and Ghahramani, 2016).

Alternatively, the reviewer's confusion might arise due to the term 'sister prediction', which seems to be used incorrectly by Althoff et al. (2021). This term was not used in the original papers on dropout (Srivastava et al., 2014) or Monte Carlo Dropout (Gal and Ghahramani, 2016). The oldest source we could find was Liu et al. (2017): *"Sister forecasts are predictions generated from the same family of models, or sister models. While sister models maintain a similar structure, each of them is built based on different variable selection process, such as different lengths of calibration window and different group analysis settings."* If this is the working definition of "sister-predictions" then the thinned models resulting from dropout would not be sister models and the resulting forecasts are not sister forecasts.

We hope that this helps. Either way, the description of dropout in the text of our paper is correct and uses language that is in agreement with primary sources.

6. L193-4: why not use a regularized squared loss function? Is it not a standard practice to perform L2 (and potentially L1) regularization to improve LSTM performance and reduce overfitting? Was this considered? If not, why?

There are many regularization techniques for neural networks. For example, we use dropout as a regularization. This is quite common. L1 and L2 regularization can yield good results in specific contexts, but they are not as trivially applicable as one might think — for example, we did indeed try L1 and L2 regularizations in some previous experiments, but did not obtain good results. For the use of L2 regularization specifically, it is now often implicitly used/approximated in the form of weight-decay.

7. L195-6: what dataset partition was used to select optimal hyper-parameters?

As is customary, we used the training- and validation-sets as defined in the appendix. We will mention this explicitly, since— as the reviewer rightfully points out — we cannot assume that readers would know.

8. Section 2.4.3: the authors should formally define first and second order uncertainties.

We will answer this together with comment 12.

9. Table 3: it's not clear how the 'obs' data is to be used to 'contextualize' the results from the different models. More detail should be provided (perhaps with an example in the relevant section).

We are not sure what the confusion is. Statistics from the observation sampling distribution are given as a reference for statistics of model error distributions. We are not sure what details could be provided or how there could be confusion about this. Basically, variances and quantile widths mean relatively little on their own

without a point of reference, and comparing to the observation distribution shows improvement due to using the UE approaches.

10. L286-289: it would seem like a good idea for the authors to investigate how best to use the CMAL and UMAL variants for improving predictive performance at low and high (tail) flows (from the point prediction point of view), as this tends to be a major impetus for creating models with uncertainty assessment capabilities. In other words, what's the point of going through the trouble of designing these more sophisticated methods if they cannot outperform the base approach (LSTMp) when assessed on highly relevant metrics. Please don't get me wrong, I am not trying to downplay the very interesting work done by the authors, I am simply trying to help them more fully explore the merits of their work and better 'sell' their approach to a 'skeptic'.

There is likely some sort of confusion here from the reviewer. All of the reported point metrics are relatively good in comparison (see e.g. Kratzert et al., 2019). The low and high-flow metrics are however measuring a form of bias that assumes a symmetric error distribution. And, the point of these sentences is that the mean of the distributional predictions is biased in low and high flow regimes, since we can expect that the underlying distributions are indeed asymmetric (e.g. there are now flows below 0). This is a necessity: There is an inherent (not incidental) tradeoff between predicting asymmetric distributions (hydrological uncertainty is well-known to be asymmetric) and using the mean of an asymmetric prediction as a point-estimate (compare Figure I of our answers). This is a fundamental attribute of probabilities that we just wanted to point out explicitly. It's not about a difference between models (no model can fix this problem), it is a fundamental artifact of probabilistic prediction. If necessary, we can revise this sentence in the manuscript, but it is not a matter of a poorly performing model or bad metric.

Thank you. This is much appreciated.

11 but it is unclear what message the authors expect the reader to take-away from this figure. What's the relevance of this figure and why should the reader 'care' about it?

We agree with the critique.

Formally, if a variable $y$ is described by a family of distributions $y = f_0(y; \theta_1)$ that are functions of possibly multidimensional parameters $\theta_1$, we call it first order uncertainty. If the potential distribution of said parameters are estimated — that is, if the parameters are "*stochasticized*" — as in $\theta_1 = f_1(\theta_1; \theta_2)$ one calls it second order uncertainty. As usual, higher orders can be derived recursively so that $\theta_n = f_n(\theta_n; \theta_{n+1})$. First order uncertainty is related to aleatoric uncertainty, and second- or higher-order uncertainties to epistemic uncertainty (we are however in a very restricted setting here and the approaches do not necessarily disentangle the different kinds of uncertainties). Similarly, the MDNs estimate first-order uncertainties, while MCD estimates second-order uncertainty.

That said, we do not believe that a formalization is the right way to go at this point of the paper. The above formalism will be familiar in one way or another to many. We think the problem here lies in our unclear usage of language. That is, a more thoughtful description is warranted. We will thus expand the section in the following way:

> "In this experiment we want to demonstrate an avenue for studying higher-order uncertainties with CMAL. Intuitively, the distributional predictions are estimations themselves and thus subject to uncertainty. And, since the distributional predictions do already provide estimates for the prediction uncertainty we can think about the uncertainty regarding parameters and weights of the components as a second-order uncertainty. In theory even higher-order uncertainties can be thought of. Here, as already described in the method-section we use MCD on top of the CMAL approach to "stochasticize" the weights and parameters and expose the uncertainty of the estimations. Figure 11 illustrates the procedure: The upper part shows a

hydrograph with the 25%–75% quantiles and 5%–95% quantiles from CMAL. This is the main prediction. The lower plots show kernel density estimates for particular points of the hydrograph (marked in the upper part with black ovals labeled 'a', 'b' and 'c', and shown in red in the lower subplots). These three specific points represent different portions of the hydrograph with different predicted distributional shapes and are thus well suited for showcasing the technique. These kernel densities (in red) are superimposed with 25 sampled estimations derived after applying MCD on top of the CMAL model (shown in lighter tones behind the first order estimate). These densities are the MCD-perturbed estimations and thus a gauge for how second order uncertainty influences the distributional predictions."

13. Section 3.3: my understanding is that this is the time needed to make predictions with a trained model. What is the training time for the different models? Can the authors provide an example calculation for the overall run-time in Appendix A (or at the very least in their reply, it's not clear how the 365 and 174 days were calculated)? This is correct and the reviewer did indeed spot an error here. Training is much faster, since no sampling is necessary! If we have one batch-size of 256, 531 basins, 10 years, with 365 days each, we have *256 x 531 x 10 x 365* = *1938150* data-points. Here, the batch-size tells us how many can be processed in parallel. We need approximately 7 minutes per epoch — i.e. to make a prediction for each data-point — since each batch takes ~0.055 seconds to compute. For, say 30 epochs, this would yield a model training-phase of 3.5 hours. The different hyper-parameter runs can of course also be parallelized.

When using MCD for sampling, the samples are generated by repeatedly re-executing the model. In the example, we originally wanted to take 75,000 samples for the 531 basins over 10 years. This means that we would need to generate *75,000 x 531 x 10 x 365* points. For illustrative purposes we assumed a batch-size of 256, as above. For practical purposes one could of course use much larger batch-sizes (whatever fits on the GPU) at no additional cost. To get the number of days from this we compute:

$$(75{,}000 \text{ x } 531 \text{ x } 10 \text{ x } 365 \text{ x } 0.055)/(256 \text{ x } 60 \text{ x } 60 \text{ x } 24) \approx 361.46 \text{ days}$$

That is, approximately 360 days (providing a simple estimate, and accounting for errors in the computation and numerical imprecisions etc.). The 174 are obtained by replacing 0.055 with 0.026 in the above computation and rounding up.

In retrospect we should have stuck with the original 7500 (which would have yielded 36.1 and 17.4 days, respectively) that we used throughout the manuscript. We will correct this for the revised version of the manuscript. and larger batch-sizes are possible in practice.

14. Section 4: after L335 the authors may wish to very briefly summarize the adopted models and datasets used in the study before continuing with L336 onwards. This should help the interested reader with a short 'time-budget', who may only jump from the abstract to the conclusion, get a decent idea of the methods and dataset involved (the dataset being one of the key strengths of this paper).

This is a very good proposal. We will do so for the revised manuscript version.

15. Each equation in the appendices (B1, B2, etc.) should be properly cited (the rule is to cite all equations that are not developed by the authors in the paper).

We agree that all equations should be properly cited. And, we believe that we did so: The GMM mechanism stems from Bishop (1999), the UMAL from Brando et al. (2020) and MCD from Gal and Ghahramani (2016). All of these are referenced. CMAL was introduced by us, so no further references are necessary. Further, we adapted the entire notation to put the equations into the present context and make it easier to see where the approaches are similar or divergent. Also here, no further references are necessary.

**TECHNICAL CORRECTIONS**

[...]

We will adapt all proposed technical corrections. Thank you for pointing these out.

**References**

- Addor, N., & Melsen, L. A. (2019). Legacy, rather than adequacy, drives the selection of hydrological models. *Water resources research*, *55*(1), 378-390.

- Althoff, D., Rodrigues, L. N., & Bazame, H. C. (2021). Uncertainty quantification for hydrological models based on neural networks: the dropout ensemble. *Stochastic Environmental Research and Risk Assessment*, *35*(5), 1051-1067.

- Başağaoğlu, H., Chakraborty, D., & Winterle, J. (2021). Reliable Evapotranspiration Predictions with a Probabilistic Machine Learning Framework. *Water*, *13*(4), 557.

- Beven, K., & Young, P. (2013). A guide to good practice in modeling semantics for authors and referees. *Water Resources Research*, *49*(8), 5092-5098.

- Cannon, A. J. (2018). Non-crossing nonlinear regression quantiles by monotone composite quantile regression neural network, with application to rainfall extremes. *Stochastic environmental research and risk assessment*, *32*(11), 3207-3225.

- Gal, Y., & Ghahramani, Z. (2016). Dropout as a bayesian approximation: Representing model uncertainty in deep learning. In *international conference on machine learning* (pp. 1050-1059). PMLR.

- Gauch, M., Mai, J., & Lin, J. (2021). The proper care and feeding of CAMELS: How limited training data affects streamflow prediction. *Environmental Modelling & Software*, *135*, 104926.

- Li, H., Huang, G., Li, Y., Sun, J., & Gao, P. (2021). A C-Vine Copula-Based Quantile Regression Method for Streamflow Forecasting in Xiangxi River Basin, China. *Sustainability*, *13*(9), 4627.

- Liu, B., Nowotarski, J., Hong, T., & Weron, R. (2015). Probabilistic load forecasting via quantile regression averaging on sister forecasts. *IEEE Transactions on Smart Grid*, *8*(2), 730-737.

- Liu, Z., Cheng, L., Lin, K., & Cai, H. (2021). A hybrid bayesian vine model for water level prediction. *Environmental Modelling & Software*, *142*, 105075.

- Meinshausen, N., & Ridgeway, G. (2006). Quantile regression forests. *Journal of Machine Learning Research*, *7*(6).

- Papacharalampous, G., Tyralis, H., Langousis, A., Jayawardena, A. W., Sivakumar, B., Mamassis, N., ... & Koutsoyiannis, D. (2019). Probabilistic hydrological post-processing at scale: Why and how to apply machine-learning quantile regression algorithms. *Water*, *11*(10), 2126.
- Renard, B., Kavetski, D., Kuczera, G., Thyer, M., & Franks, S. W. (2010). Understanding predictive uncertainty in hydrologic modeling: The challenge of identifying input and structural errors. *Water Resources Research*, *46*(5).
- Schlosser, L., Hothorn, T., Stauffer, R., & Zeileis, A. (2019). Distributional regression forests for probabilistic precipitation forecasting in complex terrain. *Annals of Applied Statistics*, *13*(3), 1564-1589.
- Srivastava, N., Hinton, G., Krizhevsky, A., Sutskever, I., & Salakhutdinov, R. (2014). Dropout: a simple way to prevent neural networks from overfitting. *The journal of machine learning research*, *15*(1), 1929-1958.
- Vaswani, A., Shazeer, N., Parmar, N., Uszkoreit, J., Jones, L., Gomez, A. N., ... & Polosukhin, I. (2017). Attention is all you need. In *Advances in neural information processing systems* (pp. 5998-6008).
- Website: Tim Wiliams 2018 on stackoverflow: https://stackoverflow.com/questions/51483951/quantile-random-forests-from-scikit-garden-very-slow-at-making-predictions, last accessed at 22.June.2021.

---

## Author Comment (AC2)

**REVIEWER 2**

**Summary**

This paper focuses on the use of several –mostly new for hydrology– concepts and methods from the machine and deep learning fields for uncertainty quantification in rainfall-runoff modelling. Specifically, it presents a large-scale application of these concepts and methods under a new framework. This large-scale application can be used as a guide for future works wishing to apply these (or similar) concepts and methods.

**GENERAL COMMENTS**

Overall, I believe that the paper is meaningful, very interesting, and well-prepared in general terms with room for improvements.

I recommend major revisions. To my view, these revisions should (mainly but not exclusively) be made in the following key directions for the paper to reach its best possible shape:

a) Key direction #1 (for details, see specific comments #1,2): To my view, the work's background should be better covered. In fact, to my knowledge there are two very relevant published studies, additionally to the studies already included in the "Introduction" section, that use LSTMs for uncertainty assessment in hydrological modelling. Also, there are research works presenting machine learning concepts and algorithms for uncertainty assessment in hydrological modelling (e.g., for probabilistic hydrological post-processing), including some few ones that conduct large-scale benchmark experiments using data from hundreds of catchments and several machine learning models (thereby also introducing benchmark procedures,

which I agree that are very rare in the field and very important). Further, I would say that the connection with the machine and deep learning fields needs to be better highlighted as well.

We will respond to these points in specific comments below, where the reviewer gives details about which papers and methods they are referring to.

b) Key direction #2 (for details, see specific comment #3): I agree with the main point raised by the other reviewer (Dr John Quilty). To my view, only through a comparison of the four deep learning methods of the paper to other statistical and machine learning methods providing probabilistic predictions (with the latter methods playing the role of benchmarks) will the paper fully achieve its aims in terms of benchmarking. I believe that this is absolutely necessary, as (i) the paper devotes a lot of space discussing its benchmarking contribution (but easier-to-apply methods are currently missing from its contents, while they have already been exploited for probabilistic hydrological modelling) and, (ii) the paper indeed offers interesting results which would mean more to the reader if compared to the results provided by easier-to-apply statistical and machine learning methods.

The reviewer has fundamentally misunderstood our aims for benchmarking. What the reviewer suggests is antithetical to what we are trying to do in terms of advancing benchmarking. We explained this more fully in response to reviewer #1's comment, however to summarize here, the reason that we feel strongly about not doing ad hoc, one-off benchmarking is that it is generally impossible for authors to correctly implement methods that they are not experts in (and are motivated to beat). We see this consistently in papers where people benchmark against methods that we are experts in — they almost never implement these methods correctly. This is why modern scientific disciplines use community benchmarking (which has some of its own challenges). The choice to *NOT* benchmark against an ad hoc selection of ad hoc implemented methods was conscious and deliberate — doing so goes against what we are trying to do with setting up a community benchmark on an open, public community dataset. Doing what the reviewer suggests would perpetuate the exact problem that we are trying to address.

Also, we are not aware of any UE methods that are easier to apply. It is possible to design very simple UE methods around existing hydrological models, but in these cases you still have to calibrate the hydrology model, and then afterward apply some statistical procedure. Here, we do all this in one go (and we do it with the current best-performing hydrological model that has been published). All we have to do here is add a probabilistic head to the model (less than 10 lines of code), and change the loss function (a few lines of code), and then use the same training procedure that any LSTM rainfall-runoff model uses. How could there possibly be any method that is simpler than this (either conceptually, computationally, or in terms of the effort it takes to apply)?

c) Key direction #3 (for details, see specific comment #4): To my view, proper scores (see e.g., Gneiting and Raftery 2007) should necessarily be computed for assessing the issued probabilistic predictions. Currently, there is an important –from a practical point of view– aspect of this work's large-scale results that is not assessed. In fact, the selected scores cannot directly and objectively inform the forecaster-practitioner which method to prefer (and when), while proper scores can.

We will provide a more detailed answer to this comment in our answer to specific comment 4 of reviewer 2.

**SPECIFIC COMMENTS**

1) To my view, the biggest contribution of this work is that it guides the reader on how to use and combine (mostly) new deep learning concepts and methods for uncertainty assessment in hydrological modelling (type-a contribution), while the introduction of a general benchmarking framework for uncertainty assessment in hydrological modelling is (as also mentioned in the "Introduction" section) a secondary (but still important) contribution (type-b contribution). For both these types of contribution and mainly for the former one, a better coverage of the study's background is required. For instance, in lines 15 and 16 it is written that "the majority of machine learning (ML) and Deep Learning (DL) rainfall–runoff studies do not provide uncertainty estimates (e.g., Hsu et al., 1995; Kratzert et al., 2019b, 2020; Liu et al., 2020; Feng et al., 2020)". This is inarguably true; however, there are machine and deep learning rainfall-runoff studies (mostly machine learning rainfall-runoff studies) that do provide uncertainty estimates, while some of them also involve large-scale benchmarking across hundreds of catchments and also use proper scoring rules (together with more interpretable scores) to allow practical comparisons. In fact, this study is not the first one proposing and/or extensively testing machine learning algorithms for probabilistic rainfall-runoff modelling and, to my view, this should be somehow recognized in the "Introduction" section during revisions. In this latter section, information on uncertainty quantification in hydrological modelling using machine and deep learning algorithms is currently scarce, although other topics (even less relevant ones) are well-covered. Especially as regards LSTM-based methods for uncertainty quantification, to my knowledge there are two published works proposing such methods in hydrological modelling and forecasting (Zhu et al. 2020; Althoff et al. 2021). To my view, these studies should necessarily be viewed as part of this work's background.

The reviewer mentioned that there are UE benchmarking studies in hydrology using large-scale, public datasets. Alas, no examples are given. But, if any such studies exist that we did not cover, we would really like to know about them — and we would include them as reference, as we did with the other UE studies we cite.

We will include Althoff et al. (2021) in the revision — this paper was published after we completed writing this paper. It is a single-basin study (which is likely not appropriate for deep learning), but it is directly relevant. We already cited the work by Zhu et al. (2020), but we cited their primary methodological paper, not the derivative study that the reviewer mentions here.

In view of specific comment 4 by reviewer 2 it is perhaps also worth mentioning that neither of these publication reports CPRS; and Zhu et al. (2020) specifically do not make use of any strictly proper scoring.

2) Moreover, I would say that the connection with the machine and deep learning fields needs to be further highlighted for the paper to become more balanced with respect to its nature. Perhaps, this could be established by referring the reader in more places in the manuscript to the original sources of the concepts and algorithms, and by adding a few examples of research works adopting (some of) the same concepts and methods for non-hydrological applications (and possibly by highlighting features that are especially meaningful for rainfall-runoff modelling applications).

All original sources for all methods were cited. If the reviewer sees that we missed one, we will definitely fix it.

3) I should also note that I agree with the main point raised by the other reviewer (Dr John Quilty). As the paper aims to establish benchmarks and benchmark procedures for future works (and as it emphasizes its practical contribution in terms of benchmarking), it would be essential to also provide a comparison with respect to easier-to-apply methods from the statistical and machine learning fields. Such methods have already been applied in the field (mainly for probabilistic hydrological post-processing), and include (but are not limited to) the following ones: linear-in-parameters quantile regression, quantile regression forests, quantile regression neural networks and gradient boosting machine.

This specific comment is largely covered by our responses to reviewer #1. In short: As the title suggests this paper tries to establish baselines for benchmarking. As such the suggestion is not in line with our intention for the framework and the baselines.

Further, in our eyes there are no simpler UE methods available that are the ones that we propose. We cover post-processing in the paper already: We cited several seminal post-processing papers and discussed the advantages of generative probabilistic methods vs. post-processing beginning in line 20 of the manuscript. Be it as it may, we view it as more complicated than building probabilistic models directly. Quantile-regression is a form of regression and does not constitute an approach as such. Random forests and gradient boosted models are not necessarily simpler. Further, we already know that XGboost does not constitute rainfall-runoff models that are as good as LSTMs (see: Gauch, Mai, Lin; 2021). Even if we were to adopt the approach of creating our own ad hoc benchmarks (which, again, goes directly against the point we are making about community benchmarking), none of these suggestions are good examples.

4) Furthermore, in lines 94–99 it is written that "the best form metrics for comparing distributional predictions would be to use proper scoring rules, such as likelihoods (see, e.g., Gneiting and Raftery, 2007). Likelihoods, however do not exist on an absolute scale (it is generally only possible to compare likelihoods between models), which makes these difficult to interpret (although, see: Weijs et al., 2010). Additionally, these can be difficult to compute with certain types of uncertainty estimation approaches, and so are not completely general for future benchmarking studies. We therefore based the assessment of reliability on probability plots, and evaluated resolution with a set of summary statistics". However, to my view proper scores (Gneiting and Raftery 2007) should necessarily be computed in this paper, as at the moment its large-scale results cannot be directly useful to forecasters-practitioners (despite the fact that the currently computed scores provide information that could be also of interest to the reader). For example, the continuous ranked probability score—CRPS score could be computed across multiple quantiles. As these scores are indeed difficult to interpret when stated in absolute terms, in the literature they are mostly presented in relative terms by computing relative improvements offered by an algorithm with respect to another (benchmark).

Therefore, one of the compared methods could serve as a benchmark for the others, and the mean (or median) relative improvements could be computed. These computations will reveal the method that forecasters would choose among the four compared ones.

We admit that we did not emphasize the general deficiencies of the (single) metrics/statistics/distances enough in this paper. Taken alone every metric has deficiencies and assumptions underlying it. And, in general we did not want to imply that any of the proposed metrics are fixed, since it is very difficult to define a meaningfully complete set of metrics for hydrological (probabilistic) predictions — and every application will have its own unique purpose. Thus, we never wanted to attempt to define a universal set of benchmarking metrics here. As a matter of fact, we hope that the proposed metrics will be adapted, refined, exchanged and complemented as benchmarking efforts will be adopted by the community. In this way, maybe at some point, canonical metrics for UE benchmarking will emerge. But, we definitely would not dare to claim to do this. We discussed this briefly in the conclusion of the current version of the manuscript. However given the reading by the reviewer we now came to believe that we should also add a broad disclaimer when introducing the method and deepen the discussion. We will do so in the revision.

Regarding the proper scoring rules (*sensu* Gneiting and Raftey, 2007), we have to say that we purposefully did not report them. This choice was made consciously and not out of laziness or oversight. The reasoning for doing so is provided in the portion of text that the reviewer cited. And, regarding CRPS specifically, one generally distinguishes between the continuous ranked probability score (CRPS) and the continuous ranked probability skill score (often abbreviated as CRPSS or CRPS score). The former integrates over the different quantiles by construction (which might be what the reviewer tries to indicate in his statement?). The latter is the usual choice in literature for providing a more interpretable score, by using the CRPS in the same style that the NSE uses variances of point estimates. This use does however require sensible baselines, such as the ones proposed in our contribution. Our approaches conceptually allow us to evaluate the performance in terms of CRPs (and also in terms of likelihood). However, computing meaningful scores is not as simple

as the reviewer thinks: Whenever a probability density function is discretized, information is lost (see for example: Gupta et al., 2021). Intuitively, (a) if the bins are very wide, a bias is introduced because the difference between the mass of the bin and of the actual continuous distribution becomes very large; (b) if the bins are very thin, almost no data can be used to estimate its properties, which induces a large variance in the estimation. In the hydrological context specifically, small bin-widths should also be distrusted because of the inherent uncertainty of the variables. Apart from a careful choice of bin-sizes, it is therefore in practice often more appropriate to evaluate the properties of the UE approaches with regard to different metrics to derive at a nuanced, paretian evaluation (see for example: Kumar, Lia and Ma, 2020). This is what we choose to do in this paper.

Finally, we want to point out that CRPS are not common in hydrology. To substantiate this claim, we did a literature review to provide a list of publications that assess UE approaches in a hydrological context. The list is not exhaustive, but we include a larger set of topics and settings than considered in our manuscript. Notably, it includes all referenced sources by the reviewer himself, which did not report proper scoring rules. In summary, from 38 references only 5 report proper proper scores; and these also examine the performance in terms of probability plots (or metrics derived thereof) and resolution.

1. Althoff et al. (2021) primarily use point-prediction metrics for evaluation, but also percentage of coverage (related to the probability plot), average width of the uncertainty intervals (i.e., a single statistic for the resolution), and average interval score (a proper scoring rule, but nut CRPRS). They also use a set of ad-hoc tests to get an intuition about the uncertainty estimation capacity.
2. Abbasour et al. (2015) report the performance in terms of point-metrics, and show the 95% quantiles visually within hydrographs.
3. Ajami et al. (2007) count the number of observations within the 95% prediction interval (i.e. a single point on the probability plot) and show visual evidence of their approach in the form of hydrographs.
4. Berthet et al (2020) report CRPS scores, but since they found them so uninformative they also evaluate in terms of the probability-plot, the sharpness and point-metrics (as we do).

5. Beven (1993) shows bounds for specific events.

6. Beven and Binley (2014) show the 90% quantiles for individual events.

7. Beven and Smith (2015) report the event-based coverage of the 95% prediction intervals (related to the probability plot).

8. Bogner and Pappenberger (2011) report metrics for point-predictions, CRPS scores (which they just interpret in terms of accuracy), and probability plots (together with derived summary statistics).

9. Coxon et al (2015) evaluate the uncertainty in terms of bound-distances (related to precision statistics) and report the performance of the point-estimates.

10. Dogulu et al. (2015) report UE estimation performance in terms of prediction interval coverage probability (related to the probability plot), mean prediction interval (related to the precision statistics), and average relative interval length related to the precision statistics).

11. Gopalan et al. (2019) report point-prediction metrics, p-factors (related to coverage, thus to the probability plot) and simulation uncertainty indices (an improper score, related to the width of the quantiles).

12. Huart and Mailhot (2008) show hydrographs and dotty-plots.

13. Kavetski et al. (2006) report RMSE and standard deviation and show the prediction bounds visually using hydrographs.

14. Liu et al (2005) show dotty plots, and provide a visual inspection for specific events.

15. Kim et al. (2020) report CRPS scores, together with probability plots and rank histograms.

16. Mantovan and Todini (2006) report percentiles of MSE values for their derived posteriors.

17. McMillan et al. (2010) use a rank histogram (similar to our deviation plot) as primary diagnostic tool. They also show exemplary hydrographs with uncertainty bounds.

18. Montanari and Koutsoyiannis (2012) report performance in terms of the probability-plot and show some exemplary hydrographs.

19. Murphy and Winkler (1984) discuss the utility and usage of the probability plot for weather forecasting. They are one of the earliest sources that we are aware of.

20. Mustafa et al. (2019) do not evaluate the predictive uncertainty as such.

21. Papacharalampous, Tyralis, and Langousis et al. (2019) use proper scores, but only report their relative performance with respect to an arbitrary benchmark model (for the sake of clarity). They also report reliability scores (related to the probability plot), the average width of the prediction interval (related to our precision statistics).

22. Schoups and Vrugt (2010) report the UE performance in terms of probability plots.

23. Shrestha and Solomatine (2008) report interval coverage probability (related to the probability plot) and mean prediction interval (related to our precision statistics). They use a derivative of the probability plot to relate model error with probability of occurrence, as well as the model residuals in dependence of the input variables.

24. Shrestha and Solomatine (2009) use a probability plot and the mean size of the prediction intervals. They also show cumulative densities for specific events.

25. Shrestha, Kayastha, and Solomatine (2009) report interval coverage probability (related to the probability plot) and mean prediction interval (related to our precision statistics). They also show the 90% prediction intervals for specific events.

26. Shortridge, Guikema, and Zaitchik (2016) do not use explicit diagnostics to assess the UE estimation capacity, but use a scenario based approach.

27. Srivastav, Sudheer, and Chaubey (2007) show the quantiles within hydrographs for specific events.

28. Teweldebrhan, Burkhart, and Schuler (2018) report point-metrics, critical success index (similar to coverage ratio, thus related to the probability plot) and show some exemplary hydrographs.

29. Tian et al. (2018) report point-estimation metrics, average size of the uncertainty interval (related to our precision statistics) and the coverage ratio differentUE quantiles (related to the probability plot).

30. Thyer et al. (2009) report the uncertainty estimation performance in terms of the probability plots.

31. Tolson and Shoemaker (2008) show prediction bounds explicitly.

32. Vrugt et al. (2005) report point-estimation metrics together with error distribution plots.

33. Vrugt et al. (2008) report point-estimation metrics, show the obtained bounds over the training and validation periods, and the standard deviation of the estimated parameters.

34. Woldemeskel et al. (2018) report CRPSS (a scoring rule), a probability plot derivative, and the 99% interquartile range (related to our precision statistics). The authors also show the 98% quantiles for specific events.

35. Westerberg and McMillan (2015) show individual runs, and quantile deviations (similar in kind to our deviation plot).

36. Zink et al. (2017) report the coefficient of variation and normalized 5%-95% quantile ranges (both related to our precision statistics).

37. Zhu et al. (2020) primarily use point-prediction metrics and visually inspect the uncertainty estimation capacities of the model.

38. Vaysse and Lagacherie (2017) report point-prediction metrics in conjunction with probability plots.

5) Also, my general feeling is that the type-b contribution of the paper (see specific comment #1) is emphasized somewhat more than its type-a contribution (see again specific comment #1) throughout the paper. To my view, the opposite would be more befitting to the contents of the paper. In any case, the type-a contribution could at least be further discussed in the "Conclusions and Outlook" section.

Since the reviewer gave no reason or explanation as to why they hold this opinion it is difficult for us to decide how (or whether) to act on this comment. What content is missing from the conclusions and outlook section? Just asking to "add more" is not helpful.

6) Moreover, the following lines (and other similar statements) do not describe the literature accurately (as some existing works on uncertainty assessment in hydrological modelling and forecasting offer benchmarks and benchmarking procedures; see also specific comment #1) and could be rephrased a bit (or removed) to recognize the relevant work made so far in the field:

1. … "while standardized community benchmarks are becoming an increasingly important part of hydrological model development and research, similar tools for benchmarking uncertainty estimation are lacking" (lines 3 and 4).
2. "We struggled with finding suitable benchmarks for the DL uncertainty estimation approaches explored here" (lines 51 and 52).
3. "Note that from the references above only Berthet et al. (2020) focused on benchmarking uncertainty estimation strategies, and then only for assessing postprocessing approaches" (lines 55–57).
4. "However, as of now, there is no way to assess different uncertainty estimation strategies for general or particular setups" (lines 332 and 333).

To our knowledge, all of the above quoted statements from the paper are correct. The reviewer provides no references that do any of these things — the reviewer only asserts that such references exist. We looked extensively for such references and found none.

The requirements for a suitable, standardized benchmark are (Nearing et al., 2018): (i) that the benchmark uses a community-standard data set that is publicly available, (ii) the model or method is applied in a way that conforms to community standards of practice for that data set (e.g., standard train/test splits), and (iii) that the results of the standardized benchmark runs are publicly available. To these we added a post-hoc model examination step in our framework, which aims at exposing the intrinsic properties of the model. Although this last step is important, especially for ML approaches and imperfect approximations, we do not view it as a requirement for benchmarking in general (and therefore would have included any paper that did items i-iii but not this).

We spent considerable time searching for such UE benchmarks for this paper. We do not believe that we "misrepresented the current research landscape" and we wrote

the quoted sentences in good faith. As a matter of fact, the difficulty to find such benchmarks was a reason why we decided to include a focus on establishing a community UE benchmark in the first place.

7) Lastly, to my view the same holds for the following lines, as there are research works using machine learning ensembles for uncertainty quantification in hydrological modelling:

"A perhaps self-evident example for the potential of improvements are ensembles: Kratzert et al. (2019b) showed the benefit of LSTM ensembles for single-point predictions, and we believe that similar approaches could be developed for uncertainty estimation" (lines 367–369)

This text passage does not propose to estimate uncertainty by using ensembles. It proposes to build ensembles of uncertainty estimators. We are not aware of any publication in the hydrological sector that has done this so far. If a reference had been provided we could cite it in this context, however we are unaware of such a study.

**References**

- Abbaspour, K. C., Rouholahnejad, E., Vaghefi, Srinivasan, R., Yang, H., & Kløve, B. (2015). A continental-scale hydrology and water quality model for Europe: Calibration and uncertainty of a high-resolution large-scale SWAT model. *Journal of Hydrology*, *524*, 733-752.
- Althoff, D., Rodrigues, L. N., & Bazame, H. C. (2021). Uncertainty quantification for hydrological models based on neural networks: the dropout ensemble. *Stochastic Environmental Research and Risk Assessment*, *35*(5), 1051-1067.
- Ajami, N. K., Duan, Q., & Sorooshian, S. (2007). An integrated hydrologic Bayesian multimodel combination framework: Confronting input, parameter, and model structural uncertainty in hydrologic prediction. *Water resources research*, *43*(1).

- Berthet, L., Bourgin, F., Perrin, C., Viatgé, J., Marty, R., & Piotte, O. (2020). A crash-testing framework for predictive uncertainty assessment when forecasting high flows in an extrapolation context. *Hydrology and Earth System Sciences*, *24*(4), 2017-2041.

- Beven, K. (1993). Prophecy, reality and uncertainty in distributed hydrological modelling. *Advances in water resources*, *16*(1), 41-51.

- Beven, K., & Binley, A. (2014). GLUE: 20 years on. *Hydrological processes*, *28*(24), 5897-5918.

- Beven, K., & Smith, P. (2015). Concepts of information content and likelihood in parameter calibration for hydrological simulation models. *Journal of Hydrologic Engineering*, *20*(1), A4014010.

- Bogner, K., & Pappenberger, F. (2011). Multiscale error analysis, correction, and predictive uncertainty estimation in a flood forecasting system. *Water Resources Research*, *47*(7).

- Coxon, G., Freer, J., Westerberg, I. K., Wagener, T., Woods, R., & Smith, P. J. (2015). A novel framework for discharge uncertainty quantification applied to 500 UK gauging stations. *Water resources research*, *51*(7), 5531-5546.

- Dogulu, N., López López, P., Solomatine, D. P., Weerts, A. H., & Shrestha, D. L. (2015). Estimation of predictive hydrologic uncertainty using the quantile regression and UNEEC methods and their comparison on contrasting catchments. *Hydrology and Earth System Sciences*, *19*(7), 3181-3201.

- Gopalan, S. P., Kawamura, A., Amaguchi, H., Takasaki, T., & Azhikodan, G. (2019). A bootstrap approach for the parameter uncertainty of an urban-specific rainfall-runoff model. *Journal of Hydrology*, *579*, 124195.

- Huard, D., & Mailhot, A. (2008). Calibration of hydrological model GR2M using Bayesian uncertainty analysis. *Water Resources Research*, *44*(2).

- Kavetski, D., Kuczera, G., & Franks, S. W. (2006). Bayesian analysis of input uncertainty in hydrological modeling: 2. Application. *Water resources research*, *42*(3).

- Kim, T., Shin, J. Y., Kim, H., & Heo, J. H. (2020). Ensemble-Based Neural Network Modeling for Hydrologic Forecasts: Addressing Uncertainty in the Model Structure and Input Variable Selection. *Water Resources Research*, *56*(6), e2019WR026262.

- Liu, Z., Martina, M. L., & Todini, E. (2005). Flood forecasting using a fully distributed model: application of the TOPKAPI model to the Upper Xixian Catchment. *Hydrology and Earth System Sciences*, *9*(4), 347-364.

- Mantovan, P., & Todini, E. (2006). Hydrological forecasting uncertainty assessment: Incoherence of the GLUE methodology. *Journal of hydrology*, *330*(1-2), 368-381.

- McMillan, H., Freer, J., Pappenberger, F., Krueger, T., & Clark, M. (2010). Impacts of uncertain river flow data on rainfall-runoff model calibration and discharge predictions. *Hydrological Processes: An International Journal*, *24*(10), 1270-1284.

- Montanari, A., & Koutsoyiannis, D. (2012). A blueprint for process-based modeling of uncertain hydrological systems. *Water Resources Research*, *48*(9).

- Murphy, A. H., & Winkler, R. L. (1984). Probability forecasting in meteorology. *Journal of the American Statistical Association*, *79*(387), 489-500.

- Mustafa, S. M. T., Hasan, M. M., Saha, A. K., Rannu, R. P., Uytven, E. V., Willems, P., & Huysmans, M. (2019). Multi-model approach to quantify groundwater-level prediction uncertainty using an ensemble of global climate models and multiple abstraction scenarios. *Hydrology and Earth System Sciences*, *23*(5), 2279-2303.

- Nearing, G. S., Ruddell, B. L., Clark, M. P., Nijssen, B., & Peters-Lidard, C. (2018). Benchmarking and process diagnostics of land models. *Journal of Hydrometeorology*, *19*(11), 1835-1852.

- Papacharalampous, G., Tyralis, H., Langousis, A., Jayawardena, A. W., Sivakumar, B., Mamassis, N., ... & Koutsoyiannis, D. (2019). Probabilistic hydrological post-processing at scale: Why and how to apply machine-learning quantile regression algorithms. *Water*, *11*(10), 2126.

- Schoups, G., & Vrugt, J. A. (2010). A formal likelihood function for parameter and predictive inference of hydrologic models with correlated, heteroscedastic, and non-Gaussian errors. *Water Resources Research*, *46*(10).

- Shrestha, D. L., & Solomatine, D. P. (2008). Data-driven approaches for estimating uncertainty in rainfall-runoff modelling. *International Journal of River Basin Management*, *6*(2), 109-122.

- Shrestha, D. L., Kayastha, N., & Solomatine, D. P. (2009). A novel approach to

parameter uncertainty analysis of hydrological models using neural networks. *Hydrology and Earth System Sciences*, *13*(7), 1235-1248.

- Shortridge, J. E., Guikema, S. D., & Zaitchik, B. F. (2016). Machine learning methods for empirical streamflow simulation: a comparison of model accuracy, interpretability, and uncertainty in seasonal watersheds. *Hydrology and Earth System Sciences*, *20*(7), 2611-2628.

- Solomatine, D. P., & Shrestha, D. L. (2009). A novel method to estimate model uncertainty using machine learning techniques. *Water Resources Research*, *45*(12).

- Srivastav, R. K., Sudheer, K. P., & Chaubey, I. (2007). A simplified approach to quantifying predictive and parametric uncertainty in artificial neural network hydrologic models. *Water Resources Research*, *43*(10).

- Teweldebrhan, A. T., Burkhart, J. F., & Schuler, T. V. (2018). Parameter uncertainty analysis for an operational hydrological model using residual-based and limits of acceptability approaches. *Hydrology and Earth System Sciences*, *22*(9), 5021-5039.

- Thyer, M., Renard, B., Kavetski, D., Kuczera, G., Franks, S. W., & Srikanthan, S. (2009). Critical evaluation of parameter consistency and predictive uncertainty in hydrological modeling: A case study using Bayesian total error analysis. *Water Resources Research*, *45*(12).

- Tian, Y., Xu, Y. P., Yang, Z., Wang, G., & Zhu, Q. (2018). Integration of a parsimonious hydrological model with recurrent neural networks for improved streamflow forecasting. *Water*, *10*(11), 1655.

- Tolson, B. A., & Shoemaker, C. A. (2008). Efficient prediction uncertainty approximation in the calibration of environmental simulation models. *Water Resources Research*, *44*(4).

- Vaysse, K., & Lagacherie, P. (2017). Using quantile regression forest to estimate uncertainty of digital soil mapping products. *Geoderma*, *291*, 55-64.

- Vrugt, J. A., Diks, C. G., Gupta, H. V., Bouten, W., & Verstraten, J. M. (2005). Improved treatment of uncertainty in hydrologic modeling: Combining the strengths of global optimization and data assimilation. *Water resources research*, *41*(1).

- Vrugt, J. A., Ter Braak, C. J., Clark, M. P., Hyman, J. M., & Robinson, B. A.

(2008). Treatment of input uncertainty in hydrologic modeling: Doing hydrology backward with Markov chain Monte Carlo simulation. *Water Resources Research*, *44*(12).

- Westerberg, I. K., & McMillan, H. K. (2015). Uncertainty in hydrological signatures. *Hydrology and Earth System Sciences*, *19*(9), 3951-3968.
- Woldemeskel, F., McInerney, D., Lerat, J., Thyer, M., Kavetski, D., Shin, D., ... & Kuczera, G. (2018). Evaluating post-processing approaches for monthly and seasonal streamflow forecasts. *Hydrology and Earth System Sciences*, *22*(12), 6257-6278.
- Zink, M., Kumar, R., Cuntz, M., & Samaniego, L. (2017). A high-resolution dataset of water fluxes and states for Germany accounting for parametric uncertainty. *Hydrology and Earth System Sciences*, *21*(3), 1769-1790.
- Zhu, S., Luo, X., Yuan, X., & Xu, Z. (2020). An improved long short-term memory network for streamflow forecasting in the upper Yangtze River. *Stochastic Environmental Research and Risk Assessment*, *34*(9), 1313-1329.

---

## Author Comment (AC3)

**REVIEWER 3: Anna Sikorska-Senoner**

**GENERAL COMMENTS**

This paper proposes a novel method for benchmarking uncertainty in river flow simulations via using novel deep learning (DL) methods and an extensive sample of 531 catchments. The manuscript is generally well written and structured and it is of a value for hydrological community and HESS readers. The great value of this work is a combination of a large sample study with novel deep learning methods for benchmarking uncertainty in rainfall-runoff models. Nevertheless, some issues as described below should be addressed before possible publication. Thus, my recommendation is a moderate to major revision.

**SPECIFIC COMMENTS**

1. The authors based their analysis on a large sample of CAMELS catchments (subset of 531 catchments), which gives a great potential for the analysis they are conducting. Thus, I found it a bit disappointing to see the results of the analysis reported only as averaged values (i.e. averaged over all catchments). I think the usage of such a large sample together with novel DL methods here applied creates a great potential to present their results in a bit more detailed way. For instance, evaluation metrics or probability plots could be presented not only for the averaged values but also giving some sample details. One way could be to present ensemble of probability plots or some ranges to give a reader a better feeling about the individual catchments' results. In a similar way, tabular values could be presented for some ranges and not only for averaged values.

Adding variance to the evaluation table is a great idea and we will include it in the revised manuscript.

Regarding the suggestion to include an ensemble of probability plots, Figure 9 in the original manuscript shows almost the same information: it shows the distributions of the results for the different quantiles over the different basins. This plot actually provides a *more* detailed look at the different quintiles and basins than an ensemble QQ plot would. We did actually try an ensemble of probability plots (before the original submission) and packing ranges or all solutions directly into the probability plot was more confusing than helpful, which is why we decided to show the results over the different quantiles and as deviations from the 1:1 line in the form of densities (which makes it much easier and comparable than point clouds or line-plots). Anyway, it is very difficult to present results for 531 basins in a constructive way, but the paper does already include almost exactly the information that the reviewer requested.

2. It is not quite clear, which period the reported values of results for four tested models referred to. Ideally, values and plots could be presented for all three periods, i.e., for training, validation and test periods with sufficient details (see comment #1).

Thank you for pointing this out. As is convention for DL/ML approaches, we only report the test period. Training and validation periods can exhibit arbitrarily good performance, thus it is generally discouraged to report the model performance there. We will add a statement about all statistics (except hypertuning) being from the test period to all figures, tables and the textual description to make this clear for readers.

3. The method section is very well written. However it provides mostly details from a single catchment perspective. Some additional details for a large sample study, as used here, would be very useful, specifically for readers without sufficient background in the methods applied here.

Thank you for this compliment. Alas, it might hint at a deficiency in our description as no part of the method section was written with a single-catchment perspective in mind. That is, the distributional predictions are certainly made for each basin and time-steps, but the DL based model as such is and should not be trained on the basis of individual basins. As is stated in line 191, all (training) data from all 531 catchments were used to train each model. Training a model per-basin would yield bad solutions.

4. Finally, I agree with both previous reviewers that a comparison to other simpler data-driven model(s) would be very useful for assessing the methods presented here. At the current stage, one can only see which method among four tested performs best. However it is difficult to judge their overall value as a comparison to simpler methods is missing. Such analysis would also add a value to the "Conclusions and Outlook" section.

Please see our answer to John Quilty's general comments and his specific comment 4.

**MINOR COMMENTS**

Figure 1: make clear whether the figure presents all CAMELS catchments or the subset you used in this study.

Thank you. The figure shows the entire CAMELS dataset. We will make sure that this becomes clear in the revised manuscript version.

Figure 2: add a & b in the figure caption for a higher readability.

Good idea, we will do this.

Table 1: remove the index a with its notation as it duplicates information from the figure caption.

It does not, but we are happy to move the information to the table caption nevertheless.

Figure 7: what is 'clipping' here? It is also not quite clear what m and n refer to. Maybe it would be easier to present figure as a scheme, when example is given for a basin 1, 2, … and then n=531. Also it should be: "In total we have 531 basins….". Add t to "For each time step t we…"

Clipping here means that samples that are below zero are set to zero. We will mention this in the figure description and weave in your suggestions.

Line 201: why do you take 7500 samples and not any other number?

The number itself is however not crucial here. We tested different cutoff-points for the sampling during the preparation of the manuscript, both by sampling different amounts of points and by using a Gaussian simulation (so that we can control the

actual underlying uncertainty. This way we found that at around 5000 points the evaluation was relatively stable. To this we added 2500 points as a margin of safety and thus obtained the 7500. The number might thus be seen as a compromise between a relatively small number of samples provided and a relatively stable statistical estimation that can be derived from the samples.

Table 3: Text "a) All metrics are computed for the samples of each timestep and then averaged over time and basins." could be removed as it is already mentioned in the table caption.

We will remove it. Our understanding is that table captions should not present new information (except about how to read the table).

Table 4: for which period are these values presented?

We would only ever report values for the test period. No paper should ever, under any circumstances, report values for training periods, unless there is a particular and clearly stated reason. We will mention this in the table caption, but it is redundant with strict rules of practice.

Figure 10: the figure presents an example of an event of some catchment. Maybe it could be useful to pick up one catchment as an example and provide detailed results for this catchment from probability plots to events.

Albeit interesting, that would be a post-hoc model examination with a different goal. We do not see how it contributes at this point.

Conclusions and Outlook: as there is no discussion section, this part could be extended. Particularly, the discussion of obtained (averaged) results is quite vague.

This part would also benefit from comparing the tested methods to a simpler data-driven model.

We will extend the conclusions and outlook with regard to the limits of diagnostics. That said, we are not aware of simpler data-driven models that could be used in this context or would be beneficial here. The proposed approaches are quite simple (either a direct estimation of the likelihood or a sampling based approach that can be used for models that estimate the maximum likelihood) and can be used in context with all models that are differentiable and able to provide the necessary estimates. Finally, ad hoc benchmarking is antithetical to what we view as critical scientific ethics, as discussed in our responses to reviewer #1.

Line 417: remove the word 'single' which is used twice.

Will be removed.

Line 430: the expression 'the training data' is used twice.

Will be removed.

Line 438: the word 'intermediate' is used twice.

Will be removed.

---

## Referee Report (RR1)

**Review hess-2021-154-ATC2**

**TITLE**

Uncertainty Estimation with Deep Learning for Rainfall–Runoff Modelling

**RECOMMENDATION**

Accept

**REVIEWER**

John Quilty

**GENERAL COMMENTS**

The authors spent considerable effort addressing all three reviewers' comments with great care and attention to detail. The authors provided clear and substantiated reasons not to implement any of the reviewers' comments they disagreed with (e.g., the suggestion to include other simple benchmarks, such as quantile regression forests).

I am happy to recommend the paper be accepted as is.

---

## Author Response (AR3)

Dear Editor,

Thank you for your kind words and measured judgment. We agree with the assessment that the manuscript should explain our rather strict view on benchmarking and should be explained at length to make sure readers can follow. We therefore expanded the introductory paragraphs of the "data and method" section with a clearer explanation based upon your and the reviewers feedback. We also expanded the description of figure 1 to make it clearer. We believe that these changes make the exposition of our chosen flavour of benchmarking much clearer.